# The cellular and immunological dynamics of early and transitional human milk

Cas LeMaster [1], Stephen H. Pierce[1,2], Eric S. Geanes[1], Santosh Khanal[1], Staci S. Elliott[3], Allison B. Scott [3], Daniel A. Louiselle[1], Rebecca McLennan [1], Devika Maulik[4], Tamorah Lewis[3,5], Tomi Pastinen[1,5] & Todd Bradley [1,2,5,6✉]

Human milk is essential for infant nutrition and immunity, providing protection against infections and other immune-mediated diseases during the lactation period and beyond in later childhood. Milk contains a broad range of bioactive factors such as nutrients, hormones, enzymes, immunoglobulins, growth factors, cytokines, and antimicrobial factors, as well as heterogeneous populations of maternal cells. The soluble and cellular components of milk are dynamic over time to meet the needs of the growing infant. In this study, we utilize systems-approaches to define and characterize 62 analytes of the soluble component, including immunoglobulin isotypes, as well as the cellular component of human milk during the first two weeks postpartum from 36 mothers. We identify soluble immune and growth factors that are dynamic over time and could be utilized to classify milk into different phenotypic groups. We identify 24 distinct populations of both epithelial and immune cells by single-cell transcriptome analysis of 128,016 human milk cells. We found that macrophage populations have shifting inflammatory profiles during the first two weeks of lactation. This analysis provides key insights into the soluble and cellular components of human milk and serves as a substantial resource for future studies of human milk.

[1] Genomic Medicine Center, Children's Mercy Research Institute, Children's Mercy Kansas City, Kansas City, MO 64108, USA. [2] Department of Pathology and Laboratory Medicine, University of Kansas Medical Center, Kansas City, KS 66160, USA. [3] Division of Neonatology, Children's Mercy Kansas City, Kansas City, MO 64108, USA. [4] Fetal Health Center, Children's Mercy Kansas City, Kansas City, MO 64108, USA. [5] Department of Pediatrics, UMKC School of Medicine, Kansas City, MO 64108, USA. [6] Department of Pediatrics, University of Kansas Medical Center, Kansas City, KS 66160, USA. ✉email: tcbradley@cmh.edu

Human milk (HM) is not only an essential nutritional source for the growing newborn, but it is also rich in immunomodulatory factors that contribute to the development of the neonatal mucosal and systemic immune system[1,2]. Interestingly, breastfeeding during infancy has been shown to protect against chronic immune-mediated diseases such as asthma and allergic rhinitis, both of which develop later in childhood after the conclusion of breastfeeding[3]. This suggests that, in addition to protecting against infection in infancy, HM could influence the development of the neonatal immune system and imprint the state of the immune system that contributes to disease in later life. The identification of the precise factors that are critical for neonatal immunity and how they may imprint the neonatal immune system are not well defined. In addition to immunoglobulins (Igs), HM contains cytokines, growth factors, soluble receptors, cells, microbiota, enzymes, lipids, and oligosaccharides that are orally transferred to the infant and could affect neonatal immunity[1,2,4,5]. Up to this point, HM studies have investigated these components separately, in diseased states or in small numbers.

There are five major isotypes of soluble immunoglobulins that are present at different levels in peripheral blood, mucosal secretions, and tissues that are defined by differences in fragment crystallizable (Fc)-mediated effector functions[6]. Of the major Igs, IgA is the dominant subtype in HM, but there are lower levels of IgM, IgG, IgE, and IgD detected[7]. One major role of these Igs has been to protect the infant from infection by passive transfer into the neonate through breastfeeding[8]. This transfer of Igs provides mucosal immunity for the infant during a vulnerable time while its own immune system is developing. It also provides immune protection generated by maternal vaccines that the infant is not approved for receiving until later in life (for example, influenza, measles, and coronavirus disease)[9–11]. While IgA and IgM have been abundantly identified in HM, less is known about their IgG counterpart subclasses, how the levels of these isotypes compare, and how Fc binding quality compares to Igs found in serum.

Cytokines, chemokines, growth and immune factors, including vascular endothelial growth factor (VEGF), epidermal growth factor (EGF), inflammatory cytokine interleukins (IL) IL-8, IL-6, IL-1β, and chemokine C-C motifs CCL2 and CCL5 have been identified in HM and have different levels than found in human blood serum[12–14]. These pro- and anti-inflammatory molecules are passively transferred to the infant from HM and could influence the development and maturation of neonatal organs and physiological systems[14,15].

Multiple studies have shown that HM not only contains soluble bioactive factors, but also heterogeneous populations of maternal-derived epithelial, immune, and stem cell types[16–19]. One of three single-cell RNA sequencing (scRNA-seq) studies on HM suggests that the cellular component is dynamic and changes over the course of lactation[19]. The other two studies investigated mammary cells and their transcriptional signatures in HM[17,18].

Considering previous studies investigated the cellular and soluble components in HM separately, we aimed to investigate these populations together using a systems approach, during the most demanding period of breastfeeding—the first 2 weeks postpartum—when milk production is just beginning and the frequency of feeding is highest[20]. This period of feeding is also thought to have a substantial impact on the development and antigenic stimulation of the otherwise naïve infant's intestinal immune system[21]. In this study, we first investigated the soluble component, employing a multiplexed bead-based technique for the detection of antibody isotypes, cytokines, chemokines, growth factors, and other soluble analytes in samples collected at two different timepoints after birth of the infant, week 1 postpartum (2–7 days postnatal) and again at week 2 postpartum (8–16 days postnatal) from 36 different mothers. We also used scRNA-seq analysis as a method of elucidating transcriptional profiles of HM cells from different donors at week 1 and week 2. Not all samples analyzed at week 1 and week 2 were paired. We found markers of inflammation, immune signaling, and development, as well as cellular profiles and cellular subsets in HM that are dynamic over the first two weeks postpartum. We report how specific soluble and cellular factors can be clustered and identified as drivers of this shifting environment of development and inflammation.

## Results

**Dynamics of soluble immune, growth and signaling factors in human milk over the first 2 weeks postpartum.** HM samples were collected during the first (week 1) and second (week 2) week postpartum from 36 mothers (Fig. 1a, Supplementary Fig. 1a, and Supplementary Table 1). Twenty-five mothers delivered at full term (>37 weeks) and 11 mothers delivered preterm (<37 weeks; Supplementary Table 1). We determined the levels of 55 soluble immune, growth, and signaling analytes in 60 samples using multiplexed bead-based assays (Fig. 1a). Specifically, we analyzed 21 cytokines, 13 adhesion molecules, 6 chemokines, 5 adipokines, 4 enzymes, and 6 other lipocalins, growth factors, and inhibitors involved in a range of cell-signaling activities (Supplementary Table 2). Lactoferrin (LTF, 386200 pg/ml), CD14 (59597.7 pg/ml), a proliferation-inducing ligand (APRIL, 56789.6 pg/ml), osteopontin (OPN, 32498.25 pg/ml), and intercellular adhesion molecule-1 (ICAM-1, 32156.4 pg/ml) had the most abundant median values in the HM for both timepoints, whereas interferon alpha (IFNα, 1.375 pg/ml), interleukin IL-10 (1.4 pg/ml), B-cell activating factor (BAFF, 1.85 pg/ml), IL-1β (2.5 pg/ml), and IL-13 (3.7 pg/ml) were the least abundant in the population of samples (Supplementary Data 1). For visualization we normalized each component using $z$-scores to display this variation between samples and timepoints (Fig. 1b). We then performed Pearson correlations on $z$-scores of each of the analytes tested and identified analytes that had similar expression patterns (Supplementary Fig. 1b). Certain classes of cytokines and growth factors had levels that were positively correlated with one another, with an overall reduction in positive correlations occurring in the second week (Supplementary Fig. 1b). CD40L, granulocyte colony-stimulating factor (G-CSF), neuropilin 1 (NRP-1), adiponectin (ADIPOQ), adipocyte-specific secretory factor (ADSF), and stem cell growth factor beta (SCGFβ) showed strong inter-correlative relationships with other analytes in week 1 but not in week 2 (Supplementary Fig. 1b). Moreover, while cytokines were overall highly correlated with one another in both weeks, OPN and L-selectin (CD62L) showed negative correlation with most analytes in week 1 whereas cystatin C (CST3), CD14, and zinc alpha 2-glycoprotein (ZAG) showed negative correlation with many cytokines in week 2 (Supplementary Fig. 1b and Supplementary Data 1).

Overall, a heatmap display of the $z$-scores of all analytes in the HM revealed visual patterns of lower analyte concentrations in the second week (Fig. 1b). We found that 27 of the 55 soluble analytes had significant (Mann–Whitney test) changes in concentration from the first to the second week of milk production, with 26 having decreased concentrations from week 1 to week 2 postpartum (Fig. 2a). Statistical significance testing was Benjamini–Hochberg corrected for multiple comparisons made. The only analyte we measured that was significantly higher in week 2 compared with week 1 was the macrophage modulator OPN ($P$ value = 0.0079), which is often upregulated at sites of inflammation, proliferation, opsonization, and tissue remodeling (Fig. 2a)[22]. Furthermore, OPN may provide benefits to the infant in the form of immunomodulation, intestinal maturation, anti-inflammatory effects, bone

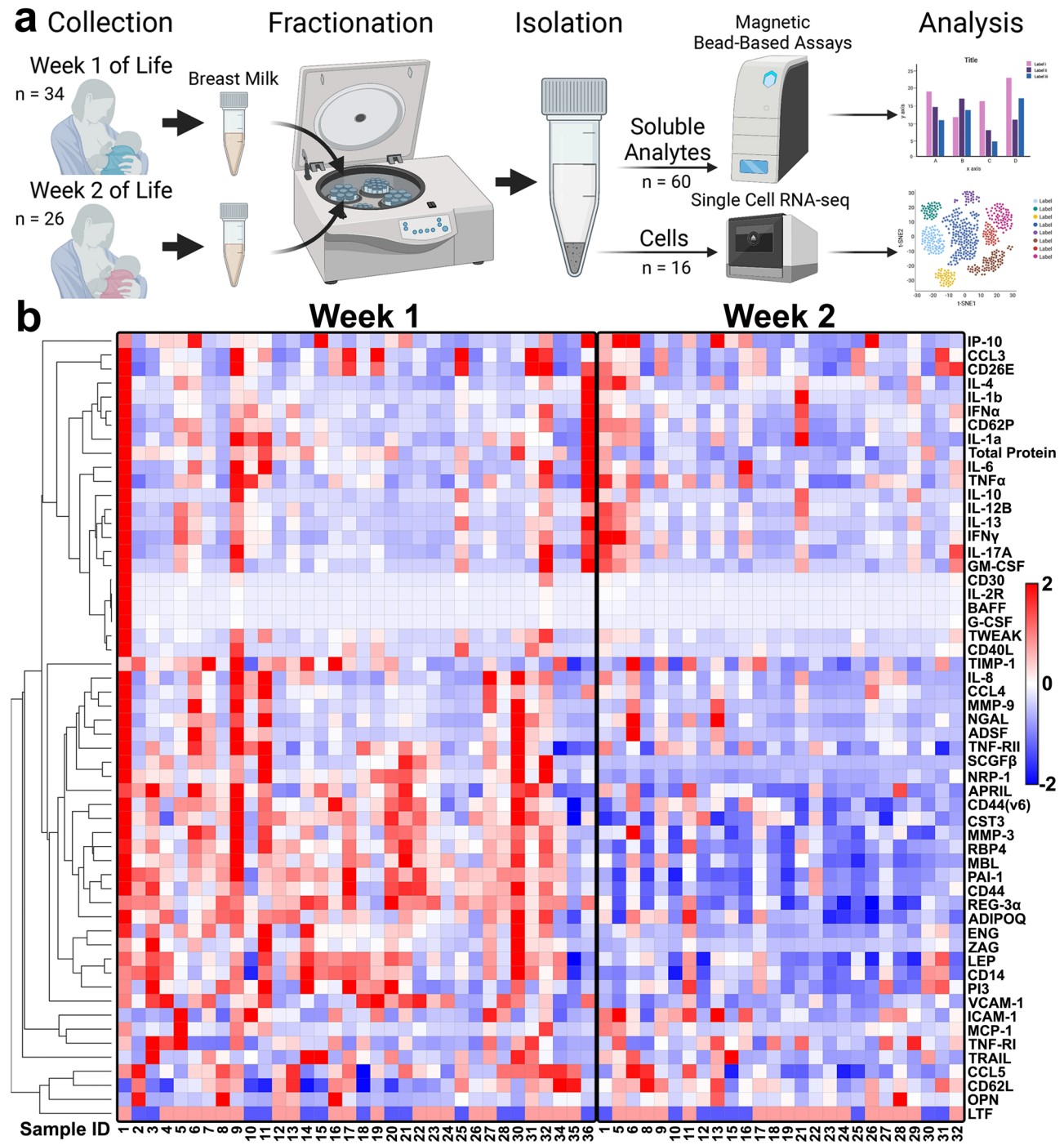

**Fig. 1 Soluble analyte concentrations in HM vary between individuals and over time. a** Schematic representation of the pipeline used to investigate the soluble and cellular components of HM, $n = 60$ samples. Created with BioRender.com. **b** Hierarchical clustered heatmap of the z-scores for 55 soluble immune, growth and signaling analytes from week 1 and week 2 HM samples ($n = 34$ and $n = 26$ samples, respectively). Sample IDs are listed under the x-axis for each week.

development, neurodevelopment, and prebiotic function[22–24]. Analytes that are involved in growth, tissue remodeling, innate immunity, inflammation, cell migration, adhesion, and metabolism, as well as total protein, had decreased levels in week 2 HM compared to week 1 (Fig. 2a). Over 22 inflammatory analytes were measured, and only 4 of the 26 analytes with significant changes were related to inflammation: chemokine ligands 3 (CCL3, P value = 0.0386) and 5 (CCL5, P value = 0.0102), tumor necrosis factor-related apoptosis-inducing ligand (TRAIL, P value = 0.0493), and APRIL (P value = 0.0079) (Fig. 2a).

Next, we compared analytes that differed in the milk from mothers that delivered preterm or full-term infants. In week 1 HM, only the B cell activating CD40 ligand (CD40L) was significantly lower in the HM from preterm infants (P value = 0.0489) (Fig. 2b). In week 2 HM, we found that CST3 (P value = 0.0489) and cellular migration proteins CD44var6 (P value = 0.0489), tissue inhibitor of metalloproteinases-1 (TIMP-1, P value = 0.0489), and CD62L (P value = 0.0489) significantly reduced in the milk from mother's who delivered preterm infants (Fig. 2c).

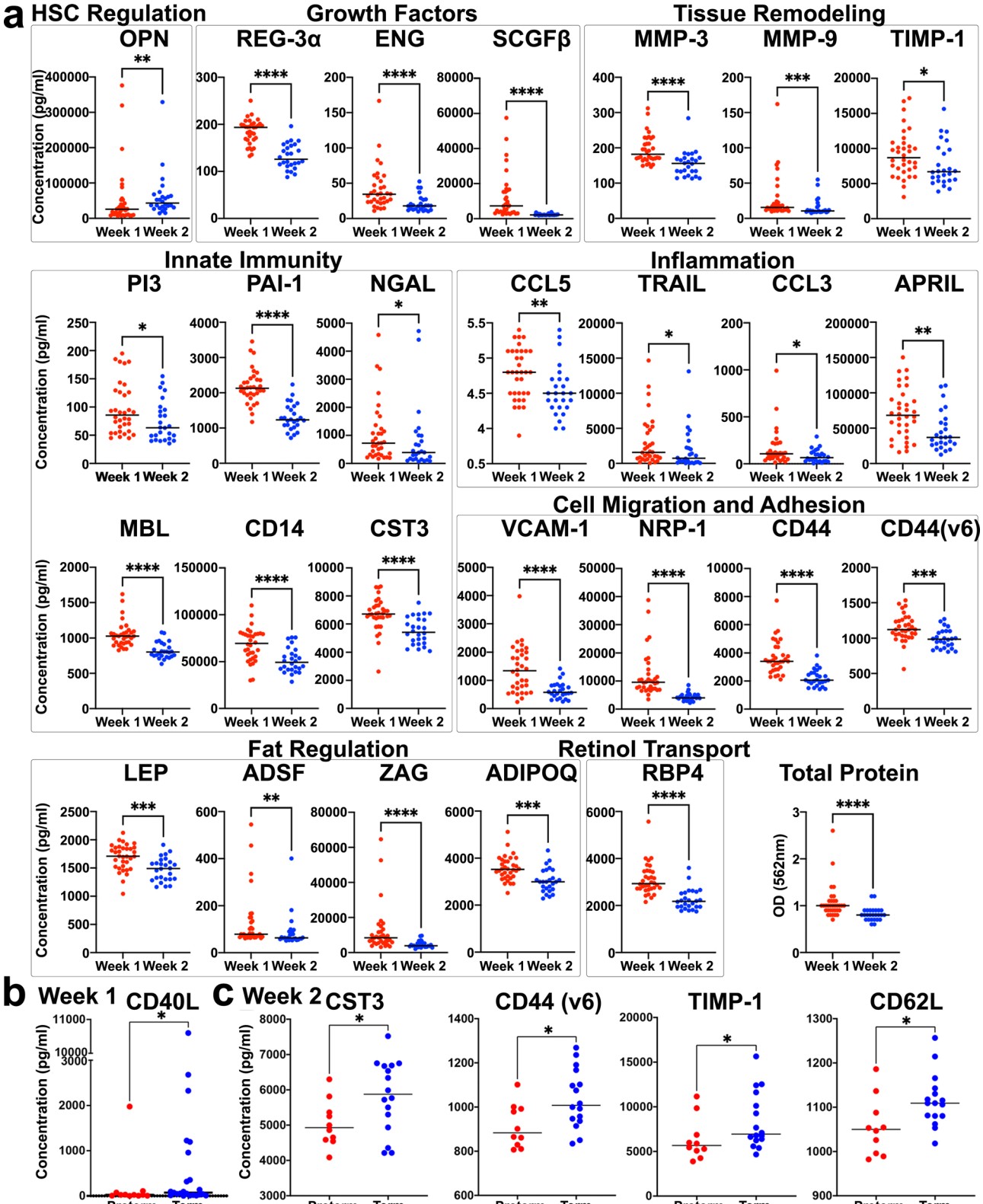

Thus, as adipokines, innate immune factors, growth factors, cell adhesion molecules, and total protein levels decrease from week 1 to week 2, most cytokines and other inflammatory analytes measured did not have significant change when comparing the first two weeks of lactation. Moreover, levels of several soluble immune cell surface receptors important for cell migration are decreased in the milk of mothers that delivered preterm infants.

**Human milk immunoglobulin subclasses differ in levels and Fc-receptor binding.** We examined the z-scores of seven subclasses of Igs in the HM samples at week 1 and week 2 postpartum for relative abundance (Supplementary Fig. 2a). As expected from prior studies of HM, IgA was detectable and within the upper and lower limits of quantitation (ULOQ, LLOQ; Median = 3546 pg/ml)[7] (Fig. 3a). Moreover, subclasses IgG1, IgG2,

**Fig. 2 Soluble analytes in HM significantly changed over the first 2 weeks postpartum and infant gestational age. a** Dot plots of soluble analytes with significant differences in concentration between week 1 (red) and week 2 (blue) ($n = 27$ of the 55 soluble analytes, 34 samples from week 1 and 26 samples from week 2). Black lines are representative of the median of all samples. Dots are representative of individual samples. Analytes with no significant changes over the two timepoints are not shown. Soluble analytes are grouped into categories based on major known functions (black box outlines). Total protein concentrations were interpolated from a standard curve. **b** Dot plot of CD40L showing significant difference in concentration between preterm (red, $n = 11$) and term (blue, $n = 23$) at week 1. Black lines are representative of the median of all samples. Dots are representative of individual samples. Analytes with no significant changes over the two timepoints are not shown. Significance was determined with Mann–Whitney tests. *$p < 0.05$. **c** Dot plots of CST3, CD44var (v6), TIMP-1, and CD62L showing significant differences in concentration between preterm (red, $n = 10$) and term (blue, $n = 16$) at week 2. Significance was determined with Mann–Whitney tests. *$P < 0.05$; **$P < 0.005$; ***$P < 0.0005$; ****$P < 0.00005$.

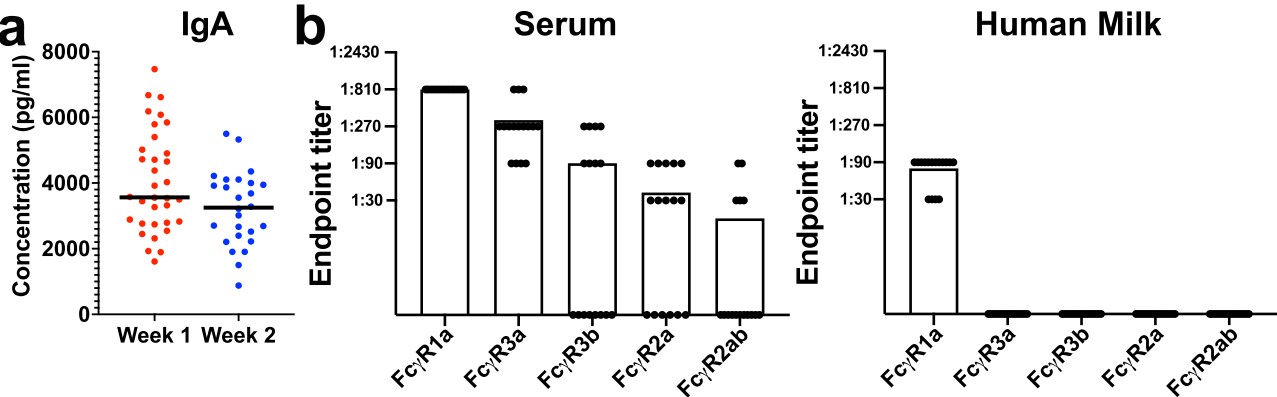

**Fig. 3 Levels of IgA in HM vary between individuals, and IgG is detectable in HM and has altered FcR binding characteristics compared to IgG from blood. a** Dot plot of IgA measured across all samples and separated by week 1 (red) and week 2 (blue) ($n = 34$ samples and 26 samples, respectively). Black lines are representative of the median of all samples at respective timepoints. Dots are representative of individual samples. Significance was determined with Wilcoxon–Mann–Whitney tests. **$P < 0.005$. **b** Bar graphs of the endpoint titers obtained by ELISA of serum and HM IgG binding to the receptors FcγR1α, FcγR3α, FcγR3b, FcγR2α, and FcγR2β at different concentrations ($n = 16$ samples and 16 HM samples). Endpoint titers are designated by the most dilute serum concentration detected above the minimum threshold of the background OD450 multiplied by 3. Dots are representative of individual samples. Bars are representative of the mean of all samples.

IgG3, IgG4, IgM, and IgE were also detected above background, but quantitative measurements could not be determined due to samples falling below the lower limit of quantitation for net median fluorescent intensity (MFI) (Supplementary Fig. 2b, c).

Because IgG subclasses of antibodies can engage multiple Fc-receptors and mediate different effector functions, we compared the binding of HM IgG to multiple Fc-receptors to those IgGs found in healthy female serum. We found that IgGs from peripheral blood could not only bind to the high-affinity receptor FcγR1α, but also had detectable binding to the low-affinity receptors FcγR3α (16/16 individuals), FcγR3β (16/16 individuals), FcγR2α (16/16 individuals), and FcγR2β (16/16 individuals) (Fig. 3b and Supplementary Fig. 3). In contrast, HM IgG could also bind the high-affinity FcγR1α, but not to any of the other FcγRs tested (Fig. 3b and Supplementary Fig. 3). This suggested that IgG antibodies from HM have different Fc-receptor binding characteristics compared to IgGs from peripheral blood and may differentially engage immune effector cells.

**Integrative analysis of soluble analytes could classify human milk into three distinct groups.** We used k-means clustering to group the HM samples to determine if concentrations of different analytes could classify HM. This revealed four distinct clusters, each with a different number of samples (Fig. 4a). Cluster 1 and 3 had similar populations ($n = 26$ samples and $n = 28$ samples, respectively), while cluster 2 had five samples (Fig. 4a). Cluster 4 had only one sample and thus a larger sample size would be required to validate if this was a true cluster or an outlier (Fig. 4a). Cluster 1 contained more week 2 samples (33.33%), whereas cluster 3 contained more week 1 samples (38.33%) (Fig. 4b).

Clusters 2 and 4 exclusively contained week 1 samples (Fig. 4b). While week postpartum influenced the cluster identity, we did not find that infant gestational age affected the cluster identify (Fig. 4c).

We performed one-versus-rest T-tests on all clusters to identify the soluble analytes that were significantly different between each of the clustered analytes. We excluded the single sample cluster 4 from further analysis. Cluster 1 possessed the most significantly different analytes, with all 19 analytes downregulated as expressed by a negative fold change (Fig. 4d). The most downregulated five analytes in this cluster are involved in cellular growth and migration (SCGFβ, NRP-1, endoglin (ENG), vascular cell adhesion molecule-1 (VCAM-1), and CD44)[25–27] (Fig. 4d). Cluster 3, however, only contained five significant analytes and all of them were slightly upregulated for cellular growth and migration (regenerating family member 3 alpha (REG-3α) and TIMP-1), anorexigenic metabolism (leptin (LEP)), and innate protection (serpin (PAI-1) and CD14)[15,28–30] (Fig. 4d). Finally, cluster 2 had 11 significant analytes and they were all upregulated more than cluster 3 analytes, with the five most elevated analytes involved in growth (SCGFβ), inflammation and immune cell recruitment (IL-8, IL-6, and IL-1α), and anti-viral activity (IFNα)[25,31] (Fig. 4d). Moreover, the most downregulated analyte in cluster 1, hematopoietic stem cell regulator SCGFβ (Adjusted $P$ value = 0.02, Log2FC = −2.11), appears to be diametrically upregulated in cluster 2 (Adjusted $P$ value = 0.001, Log2FC = 2.08) with cluster 2 insignificantly falling in the middle (Adjusted $P$ value = ns, Log2FC = −0.06) (Fig. 4d). SCGFβ is commonly found upregulated in conjunction with other cytokines promoting macrophage development[32]. Thus, levels of multiple analytes associated with immune cell function and migration could group

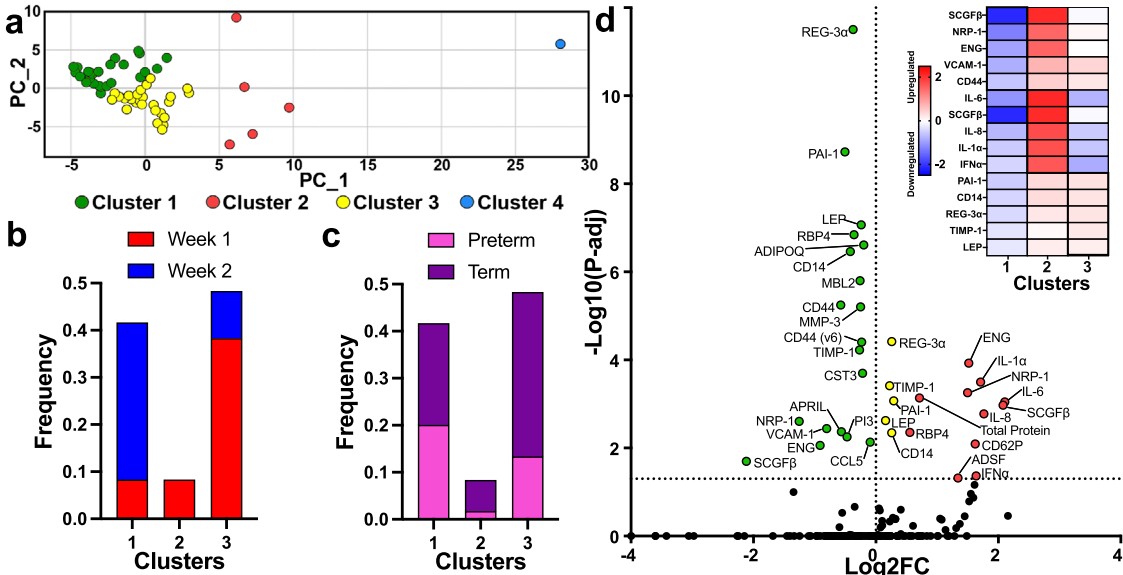

**Fig. 4 HM soluble analyte expression could cluster samples into distinct groups. a** Plot of k-means clustering, using soluble analyte concentrations, show four distinct clusters: Cluster 1 (green, $n = 26$ samples), Cluster 2 (red, $n = 5$ samples), Cluster 3 (yellow, $n = 28$ samples), Cluster 4 (blue, $n = 1$ sample). **b** Bar graph of the frequency of total week 1 (red) and week 2 (blue) samples within each cluster ($n = 60$ samples). **c** Bar graph of the total frequency of samples from preterm (<37 weeks, pink) and term (>37 weeks, purple) pregnancies within each cluster ($n = 60$ samples). **d** Volcano plot of Bonferroni adjusted $P$ values of the soluble analytes contributing to cluster formation, Cluster 1 (green), Cluster 2 (red), and Cluster 3 (yellow) dots are colorized above the $-\mathrm{Log10}(P\text{-adj})$ significance threshold. The horizontal dotted line, $y = 1.30$, represents $P = 0.05$. (Inset) Heatmap of the top 5 down or upregulated significant analytes in each cluster generated by Log2FC ($n = 59$ samples).

HM. These groupings were associated with the week in which the HM was collected but were independent of infant gestational age.

**scRNA-seq identified heterogeneous cell populations in human milk.** We performed scRNA-seq on ten week 1 and six week 2 HM samples using cells pelleted from the soluble HM portion using the 10x Genomics scRNA-seq platform (Fig. 1a and Supplementary Fig. 1a). We used Cell Ranger and Seurat to filter and analyze the scRNA-seq results. We included cells with >400 but <5000 expressed genes and excluded cells with >25% mitochondrial transcript expression (Supplementary Fig. 4a, b). This resulted in the determination of the single-cell transcriptome of 128,016 HM cells, 77,150 cells from week 1 HM samples and 50,866 cells from week 2 HM samples. First, we aggregated week 1 and week 2 HM cells together and identified 24 distinct clusters of cells using graph-based clustering of the single-cell transcriptomes (Fig. 5a and Supplementary Fig. 4c). We utilized two strategies to classify the types of cells present, gene set enrichment (GSE) by SingleR for broad cluster annotation and literature-based cell type markers for further resolution (Supplementary Table 3). Moreover, we determined the frequency of HM cells in each cluster and cluster numbers were ordered by highest to lowest HM cell frequency (Fig. 5b and Supplementary Table 3). In agreement with previous studies, the most abundant cells were lactocyte epithelial cells (LC1 and LC2), which represented 58% of total cells across eleven clusters[18,19] (Supplementary Table 3). These lactocyte epithelial clusters were enriched with milk synthesis gene lactalbumin (*LALBA*) (Fig. 5b, c). As described in previous studies, LC1 markers (Claudin 4 (*CLDN4*) and Krueppel-like factor 6 (*KLF6*)) highlighted one cluster[18,19] (Fig. 5a, b, d). Conversely, LC2 markers (xanthine dehydrogenase (*XDH*), casein alpha S1 (*CSN1S1*)) identified a larger group of cells across 10 clusters and less distinct sets of genes than LC1 clusters (Fig. 5b, d and Supplementary Data 2). Nine clusters were identified as macrophages, confirmed with elevated expression of

common macrophage marker cluster of differentiation 68 (*CD68*)[33] (Fig. 5a, e). These macrophage clusters also showed heavy monocyte enrichment, further identified by pan-monocyte marker Fc gamma receptor III alpha (*FCGR3A*) (Fig. 5a, e). T cells showed high expression in one cluster where pan-T cell marker *CD3D* was almost exclusively expressed (Fig. 5a, e). B cells were identified in one cluster and confirmed using pan-B cell marker cluster of differentiation 79 A (*CD79A*) (Fig. 5a, e). Neutrophils were located in one cluster and confirmed with known marker c-x-c motif chemokine ligand 8 (*CXCL8*)[34,35], though some expression is seen throughout other immune cell dominant clusters (Fig. 5a, e). Natural killer (NK) cells were found throughout a single cluster, the same cluster as T cells, and confirmed with natural killer cell granule protein 7 (*NKG7*) (Fig. 5a, e). Though having mixed enrichment results, cell cycling was predicted in one cluster neighboring LC2 epithelial cells, based on high expression of cell cycle marker topoisomerase II alpha (*TOP2A*) (Fig. 5a, b and Supplementary Data 2)[36].

We further investigated hematopoietic stem cells (HSC) in the HM cellular population and found 72 cells with HSC marker cluster of differentiation 34 (*CD34*). Furthermore, embryonic stem cell genes POU Class 5 Homeobox 1 (*POU5F1*), Homeobox protein NANOG (*NANOG*), and SRY-box 2 (*SOX2*) revealed 53, 29, and 0 cells, respectively. No cell expressing an embryonic or hematopoietic marker was found co-expressing another stem cell marker.

We determined the most variable genes that are significantly upregulated in each cluster (Fig. 5f). We found that LC clusters markedly expressed and differed on milk production, anti-microbial, differentiation, and growth regulating genes (Fig. 5f and Supplementary Data 2). Conversely, immune cell clusters were defined by their inflammatory, antigen processing, activation, and remodeling genes (Fig. 5f and Supplementary Data 2). Thus, we found that HM contains diverse populations of lactocytes, macrophages/monocytes, neutrophils, B and T cells, dendritic cells, and NK cells.

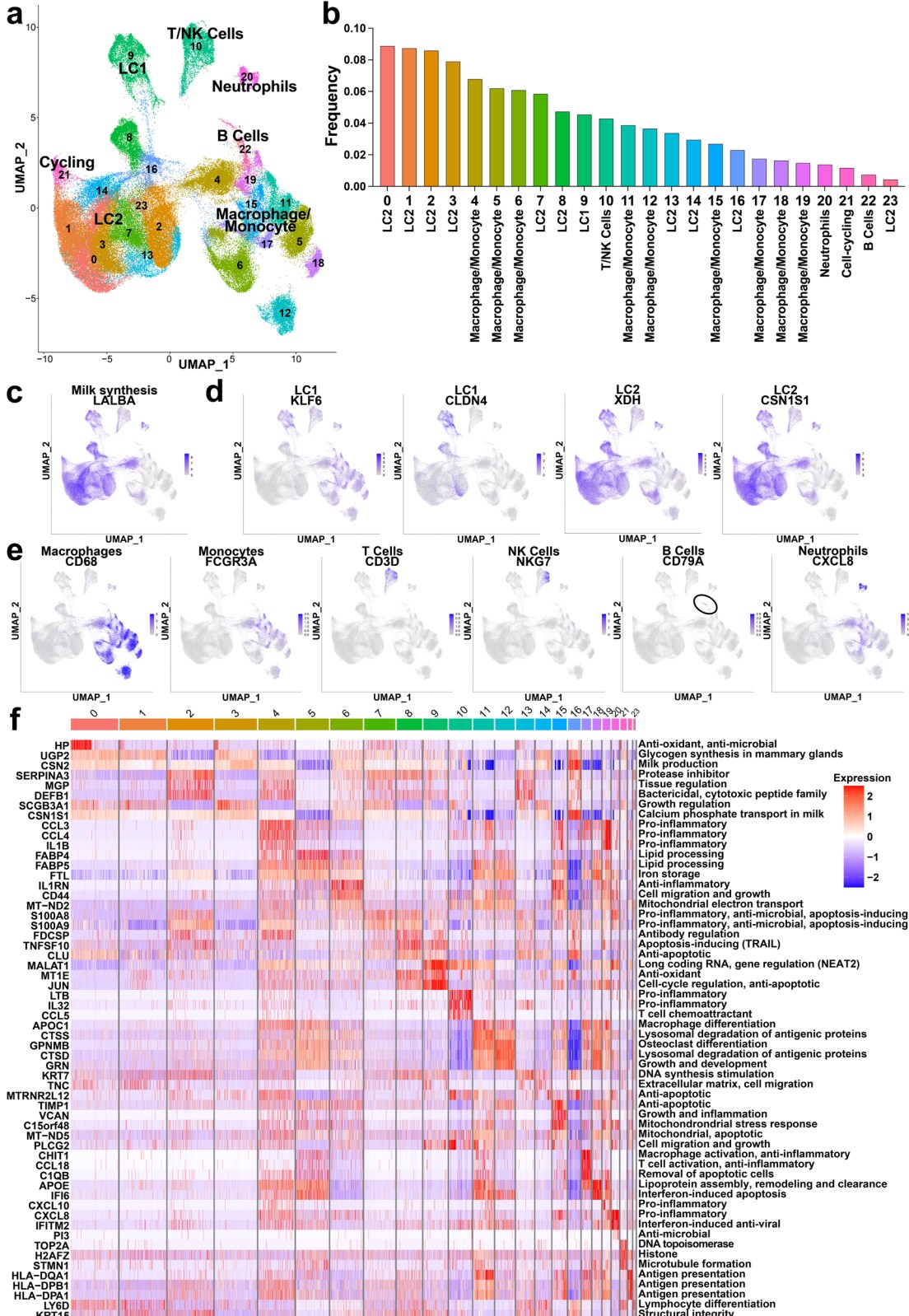

**Fig. 5 Human milk contains a diverse array of maternal cell types. a** UMAP of single-cell transcriptomes derived from HM ($n = 128{,}016$ cells). Clusters are labeled numerically (0–23) and by the dominant cell types. **b** Bar graph of the frequency of total cells within each cluster. Dominant cell types are listed below. **c** UMAP of all single-cell transcriptomes, highlighting (blue) milk synthesis gene *LALBA*. **d** UMAPs of all single-cell transcriptomes, highlighting (blue) lactocyte cells (LC) with LC1 markers (*XLF6 and CLDN4*) and LC2 markers (*XDH and CSN1S1*). **e** UMAPs of all single-cell transcriptomes, highlighting (blue) macrophage marker *CD68*, monocyte marker *FCGR3A*, T cell marker *CD3D*, NK cell marker *NKG7*, B cell marker *CD79A* (circled), and neutrophil marker *CXCL8*. **f** Heatmap of the *z*-scores based on gene expression of the top variable genes within each cluster and their main functionality (right).

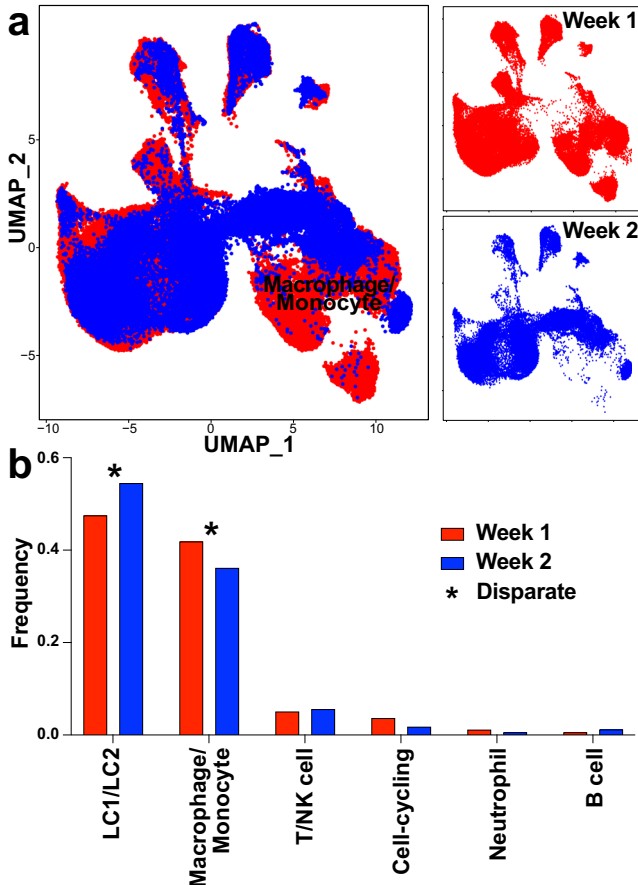

**Fig. 6 Clusters dominated by macrophages and epithelial cells differ between weeks 1 and 2. a** UMAPs of all single-cell transcriptomes, highlighting the cellular distribution from week 1 (red, $n = 77,150$ cells) and week 2 (blue, $n = 50,866$ cells) in the aggregate clustering profile. **b** Bar graph of the average frequency of cells from individuals at each timepoint. Disparate cell types have a higher frequency (>5% different) of cells from one timepoint over the other. The dominant cell type is listed below the bar.

**Monocytes/macrophages had dynamic phenotypes in early to transitional milk production.** We next compared the differences in cells from week 1 and week 2 of lactation by grouping the cells by the timepoint that the HM was collected (Fig. 6a). Using SingleR, we were able to broadly reveal changes in the frequency of human primary cell atlas (HPCA) phenotypes at each week, with an increase in epithelial cell types and neutrophils and a decrease in macrophage and B cells occurring in the second week (Supplementary Table 4). The average cell frequency at each timepoint and cell type was determined in order to identify any substantial differences in cell frequency between timepoints (Fig. 6b). We defined disparate cell types as having a frequency of at least a 5% difference between timepoints (Fig. 6b). Based on our disparity cut-off, we detected differences in frequencies between weeks for LC1/LC2 and monocyte/macrophage cells only. Thus, we found that monocyte/macrophage cells were the only immune cells with disparate frequencies between weeks postpartum (Fig. 6b).

Due to the observed shift in the cellular transcriptome of macrophage/monocyte clusters between timepoints we focused on determining differences in these cellular populations between week 1 and 2 of lactation (Figs. 6b, 7a). We first visualized the expression of the macrophage marker, *CD68*, in only week 1 cells (red) (Fig. 7b) and only week 2 cells (blue) which confirmed equivalent expression of this macrophage maker in this subset of

cells (Fig. 7c and Supplementary Table 5). We determined differential gene expression between week 1 and week 2 for these macrophage cells and identified increased anti-inflammatory genes in week 1 and increased pro-inflammatory genes in week 2 (Supplementary Data 3). Specifically, Wilcoxon tests determined the anti-inflammatory encoding genes for interleukin 1 receptor antagonist 1 (*IL1RN*, adjusted *P* value <0.0005) and transforming growth factor beta 1 (*TGFβ1*, adjusted *P* value <0.0005) were expressed in 49 and 47% of week 1 macrophage clustered cells and only 40 and 39% in week 2 with 1.14 and 0.41 log2FCs, respectively (Fig. 7d, e). Conversely, the pro-inflammatory cytokine encoding genes interleukin 1 beta (*IL1β*, adjusted *P* value <0.0005) and c-c motif chemokine ligand 4 (*CCL4*, adjusted *P* value <0.0005) were expressed in only 8 and 21% of week 1 macrophage clustered cells and 42 and 54% in week 2 with −1.73 and −1.05 log2FCs, respectively (Fig. 7f, g). We found that the macrophages in our dataset could be described regarding their inflammatory state and the observed differences in macrophage gene expression patterns reflect the inflammatory environment in HM evolving over the first 2 weeks of lactation.

**Expression of genes that encode abundant soluble protein factors in HM detected in the cellular compartment.** We sought to determine the gene expression patterns within single cells of the most abundant soluble proteins that we identified in human milk. The five most abundant soluble protein factors based on median concentration levels among all the HM samples were LTF, CD14, OPN (encoded by *SPP1* gene), APRIL (encoded by *TNFSF13*), and ICAM-1 (Fig. 1). We determined the gene expression pattern of each of the genes that encode these soluble factors across all the HM cell populations (Fig. 8). All five genes were detected in the HM cells scRNA-seq. *LTF*, *SPP1*, and *CD14* had the highest expression levels across the HM cells, whereas *TNFSF13* and *ICAM1* had lower expression (Fig. 8). *LTF*, *SPP1*, and *CD14* were most frequently detected in the mammary epithelial cells (66, 62, and 58%, respectively), but also were detected in monocyte/macrophage (28, 32, and 38% respectively) and other immune cell populations (Fig. 8). *TNFSF13*, was most frequently expressed in monocyte/macrophage cell types (51%), but also highly abundant in the mammary epithelial cells (46%). *ICAM*1 expression was more frequently detected in the immune cell populations (monocyte/macrophage, neutrophil, and B cells; 89%) with a smaller frequency of expression in the mammary epithelial cell populations (11%), although still detected in those cells (Fig. 8). This data suggested that HM cells could express the proteins that are detected in the soluble HM fraction, and some factors/genes have a cell type specific pattern of expression.

**Discussion**
In the first week postpartum, HM is composed of a mixture of colostrum and early HM. In the second week postpartum, the milk environment transitions into mature milk production. This major transitional stage in milk production affects the mother and the infant through a network of cells and bioactive factors, two dynamic components with understudied relationships. In this study, we implemented a systems approach to characterize both soluble and cellular HM compositions over the first two weeks postpartum.

The most abundant soluble analytes at both timepoints were LTF, CD14, APRIL, OPN, and ICAM-1. Many of these analytes are critical in establishing and protecting healthy guts in newborn infants. For example, LTF is known to be one of the most abundant enzymes present in colostrum and functions as a bacteriostatic agent in the intestinal tracts of infants as well as an immunomodulator by inhibiting certain inflammatory

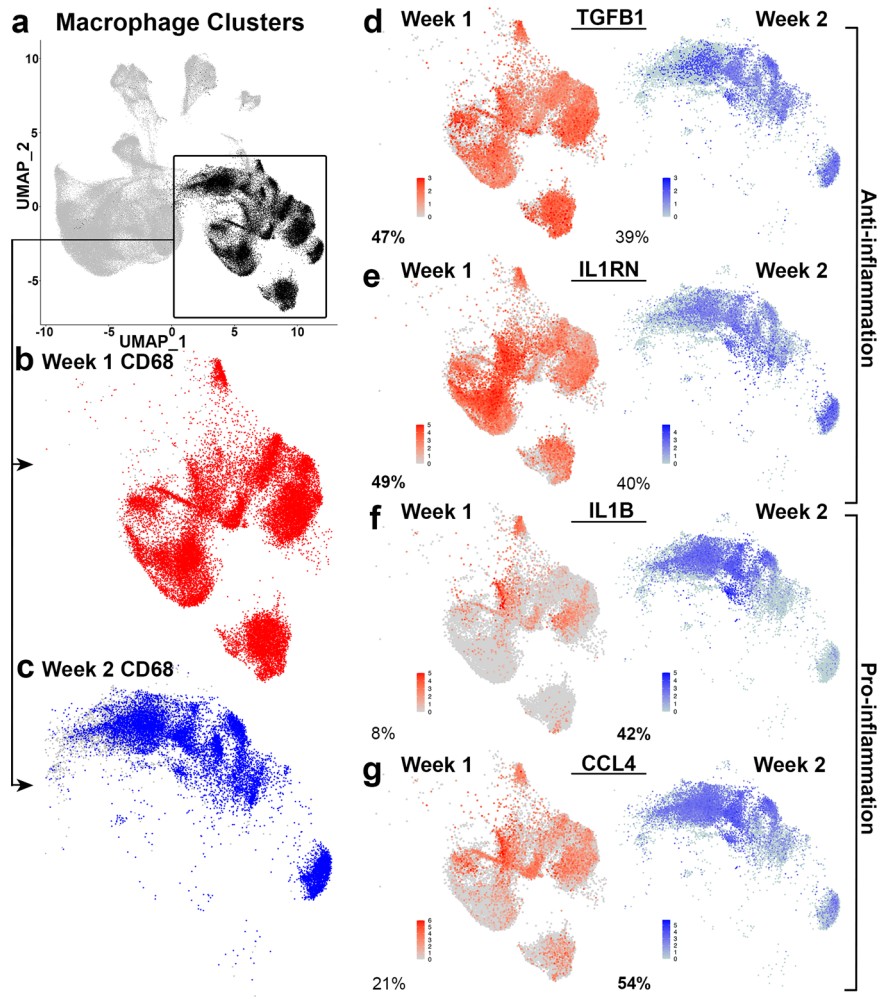

**Fig. 7 Macrophages shift from anti- to pro-inflammatory between weeks 1 and 2. a** UMAPs of all single-cell transcriptomes, highlighting of the macrophage-rich clusters with disparities between timepoints (black). **b** UMAP of macrophage-rich clusters highlighting week 1 only cells expressing macrophage marker *CD68* (red, expression >50%). **c** UMAP of macrophage-rich clusters highlighting week 2 only cells expressing macrophage marker *CD68* (blue, expression >50%). **d** UMAP of macrophage-rich clusters showing week 1 (red, 47%) and week 2 (blue, 39%) cells expressing anti-inflammatory marker *TGFβ1*. **e** UMAP of macrophage-rich clusters showing week 1 (red, 49%) and week 2 (blue, 40%) cells expressing anti-inflammatory marker *IL1RN*. **f** UMAP of macrophage-rich clusters showing week 1 (red, 8%) and week 2 (blue, 42%) cells expressing pro-inflammatory marker *IL1β*. **g** UMAP of macrophage-rich clusters showing week 1 (red, 21%) and week 2 (blue, 54%) cells expressing pro-inflammatory marker *CCL4*.

cytokines[37]. Soluble CD14, another major component of HM, has been shown to, at least partially, control innate and adaptive immune responses in the intestinal tracts of infants[38]. Finally, soluble ICAM-1 is a cell surface receptor known to inhibit infection of respiratory syncytial virus[39]. The expression of most analytes examined either decreased or remained statistically equivalent over time. However, OPN was of particular interest in the soluble component because it was the only analyte with higher expression in the second week when compared to the first. It also showed negative correlations with other analytes in the first week. OPN is often found upregulated with other cytokines and is thought to promote monocyte differentiation into macrophage[40]. Additionally, OPN may benefit infant development and maturation of gut-associated lymphoid tissue (GALT), central nervous, skeletal, and immune systems[22–24]. Our soluble analyte data suggested that as the milk environment transitions into mature milk production between the two weeks, there is a high expression of factors that can protect an infant from infections and potentially increase immune activation, including macrophage differentiation.

When comparing concentrations of soluble analytes between preterm and term gestational ages, CD40L was the only significantly different analyte from preterm to term in the first week, with preterm pregnancies having a lower concentration. CD40L is expressed on the surface of activated T cells and interacts with CD40-expressing B cells under inflammatory conditions[41]. As a soluble ligand, CD40L retains many of its functional roles, such as the promotion of cytokine production in dendritic cells and establishment of the gut mucosal inflammatory response, as well as isotype switching and immunoglobulin production[42]. Previously, colostrum was reported to express significantly higher proportions of CD40L-expressing T cells when compared to blood, suggesting a possible compensation for the lack of CD40L-expressing T cells in newborn infants and functioning as one of the mechanisms for immune protection via HM[43]. Additionally, in the second week, CD62L, which is also known as L-selectin, was one of four analytes (CST3, CD44var6, and TIMP-1) with significantly lower concentrations in preterm HM. Considering that CD62L is involved in controlling the trafficking of T cells to and from peripheral lymph nodes and is shed from the cell

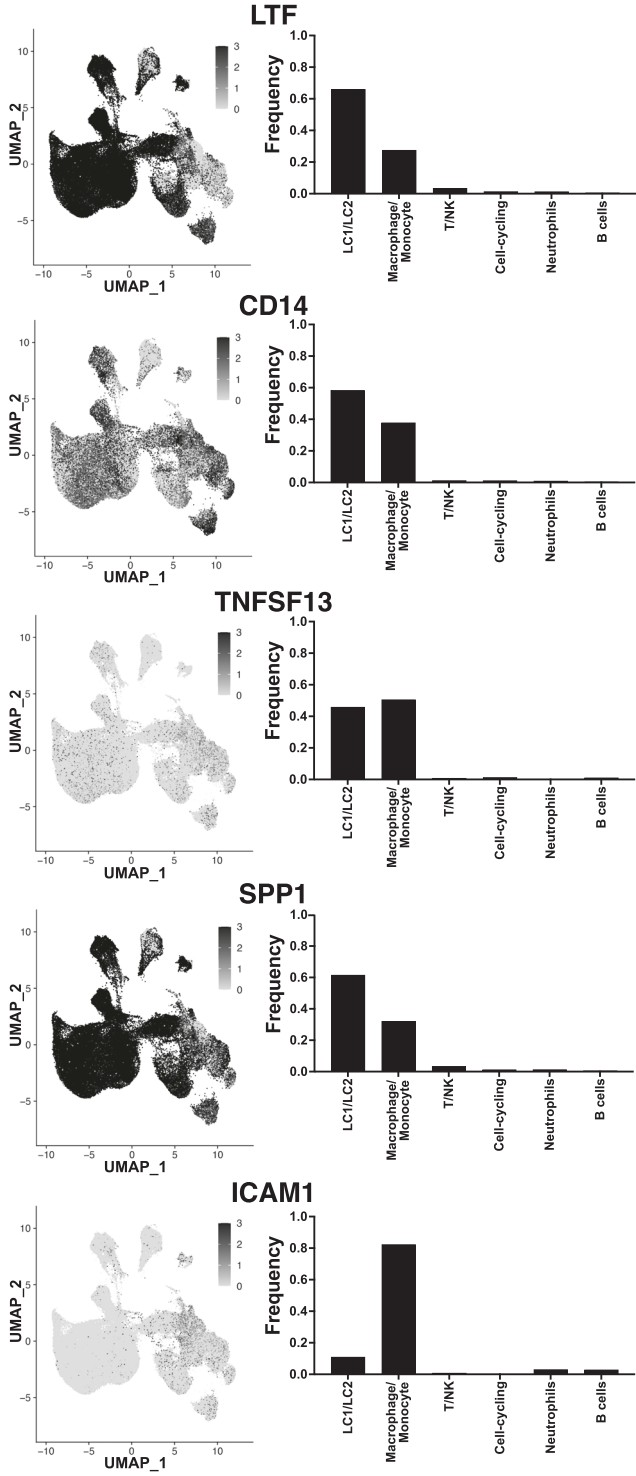

**Fig. 8 Cellular expression of genes encoding most abundant soluble analytes found in HM.** UMAP plots of the cell expression patterns of markers for genes encoding the most abundant analytes identified from the soluble HM protein analysis are displayed to the left of corresponding bar graphs showing the total frequency of the gene across cell types identified in human milk. The genes encoding soluble analytes are displayed in descending order of soluble abundance in HM.

membrane following T cell activation, together with CD40L, we see that preterm infants across the first 2 weeks postpartum may have a lower incidence of milk-derived activated T cells providing immune protection[44].

When we examined the immunological differences in the antibody repertoire of HM, we found a considerable presence of class IgA[45]. IgA is considered essential in preventing microorganisms from invading via mucous membranes and administrating anti-inflammatory tolerance to microbial and food antigens in neonates[7]. IgA has previously been found to be the most abundant antibody in colostrum and then declined in mature milk[46]. In our study, we did not find statistical changes in the concentration of IgA over the first 2 weeks. Additionally, though we did not determine quantitative measurements of other Igs in HM, we did detect them. It is likely future studies will need to employ isotype-specific dilution schemas to accurately measure HM Ig concentrations. IgM, together with IgG, is involved in the innate response, protecting mucosal surfaces from viruses and bacteria, and acting as a first line of pathogen defense[7]. Though the presence of IgG in HM has been previously shown, very little is known about the context of IgG subclasses in the protection of the neonate[7]. In the human body, however, T-dependent protein antigens elicit primarily IgG1 and IgG3 antibodies, whereas T-independent carbohydrate antigens elicit primarily IgG2 antibodies[47]. Chronic antigen stimulation, such as allergic desensitization, elicits IgG4 antibodies[47]. Moreover, evidence suggests that IgG receptors (FcγRs) comprise high-affinity and low-affinity receptors with both high-affinity and low-affinity FcγRs binding IgG-immune complexes with high efficiency; however, only high-affinity FcγRs bind monomeric IgG[47]. Here, we demonstrated that HM-derived IgGs bound to the high-affinity receptor, FcγR1α, and not low-affinity receptors. These results suggested that HM IgG has different FcR binding characteristics compared to IgGs from peripheral blood that may affect which effector cells the antibody engages. This could lead to HM IgGs engaging less neonatal effector cells in the gut and limiting inflammation. One limitation of our samples is that we did not have matched serum samples to the milk samples. Lactating women could have altered serum antibody repertoires that are different than non-lactating women. However, our observation agreed with another study that identified reduced FcγR2a and FcγR3a binding in HM compared to matched serum samples after SARS-CoV-2 infection[48]. Moreover, differences in the Fc binding patterns have also been identified in matched samples examining placental transfer to the fetus[10,49]. Our data and these studies suggest that there are differences in antibody affinity for Fc receptors that are transferred to infants via milk and placenta that could impact neonatal immunity. Future studies of what characteristics of the HM IgGs limit the FcR engagement, such as modifications like glycosylation, will be key to understanding this mechanism[50]. Additionally, processing and optimization strategies should be employed to consider the high fat and protein content of milk, when compared to serum, which may or may not impact binding efficiency.

When we analyzed how samples from mothers during week 1 and week 2 postpartum clustered together based on their soluble profile, we found 3 distinct clusters that could be defined by fold change differences in select analytes that were significantly different between clusters. One analyte, SCGFβ, consistently reflected the regulation consensus of analytes across each cluster (that is, where cluster 1 showed overall downregulation, SCGFβ was the most downregulated analyte). SCGFβ is commonly found upregulated in conjunction with other cytokines promoting macrophage development[32]. However, SCGFβ is incompletely understood and has mostly been studied in the context of stem cell therapy as a direct measure of CD34+ cells and as a growth factor in tumor cells[51]. While we did find CD34+ cells present in HM, they were in low abundance (72 cells). Moreover, the gene encoding SCGFβ, known as the C-type lectin domain containing 11A (*CLEC11A*), was not found to be differentially expressed in

our HM cell population. Previous analysis of SCGFβ in HM has not been published and may warrant further exploration. Future studies of larger sample sizes and mothers with increased demographic and health diversity could confirm and expand the clustering analysis and features we identified using soluble factors in the milk.

Previous studies have successfully traced mouse GFP+ maternal milk cells to mouse pup brain and blood destinations, illustrating the importance of mouse milk cells in early pup development[52]. Additionally, multiple scRNA-seq studies have recently shown heterogeneous populations of maternal-derived epithelial, immune, and stem cell types in HM[16–19]. However, the fate and function of human maternal milk cells are unclear in the developing neonate. Understanding the composition of milk cells and future studies of their function in the milk and when transferred to the infant will be critical. Consequently, we investigated the cellular composition in HM as the milk environment moves from early to transitional milk production. Our findings not only corroborated the major cell types found in previous single-cell RNA sequencing studies on HM, but we also expanded on them by analyzing our large dataset of HM cells[16–19]. In clustering the cells and performing differential gene expression, we discovered the largest assortment of cells as mammary epithelial phenotypes, followed by immune cells, specifically macrophages, monocytes, T cells, NK cells, neutrophils, and B cells. When determining how these cell populations evolve over the course of early lactation, we found most changes were reflected in monocyte/macrophage clusters, which reflected findings in a recent study that classified these changes as M1/2 macrophage polarizations[19]. Epithelial phenotypes have been previously explored in other studies using scRNA-seq[16–18]; however, less is known with regard to the potential roles of the macrophage populations over time. Although we attempted to classify the macrophage in our samples at each timepoint as M1/2 polarizations, we found that they could be more simply defined by their inflammatory profile, with week 1 expressing a more anti-inflammatory state and week 2 a pro-inflammatory state. Up-regulation of anti-inflammatory markers in week 1 macrophage could be an injury-like response from the expansion of breast tissue in preparation for milk synthesis[53]. Moreover, week 2 pro-inflammatory markers could indicate the recruitment of macrophages for extracellular matrix remodeling of secretory milk ducts needed in mature milk production and subsequent lipid digestion[53]. These inflammatory states were not clearly reflected in the soluble component, as soluble pro-inflammatory chemokines CCL3 and CCL5 were significantly downregulated in the second week. Cross-analysis of the products of the genes for week 2 (CCL4, IL1β) macrophage did not show significantly different concentrations across weeks. Considering that the total protein levels in the second week are significantly lower, one could deduce that an increase in select proteins during the second week might be masked by the mother's overall lower milk protein concentration.

While it was known that HM contains a heterogenous mix of leukocytes and mammary epithelial cells, less was known about the relationship these two groups have with each other early on in milk production and the soluble factors involved. A limitation of our study is that we examined timepoints separately, with several samples not having a direct pair. Moreover, we only investigated two timepoints with a wide range of time for each point. Future study designs would benefit from more frequent sampling over a longer duration of lactation. Furthermore, controlling for milk expressions, such as full or partial expression, time of day, and differentiations in milk type (i.e., colostrum), could provide better insight into the effect of breastfeeding regimens on these components. Additionally, controlling for other covariates such as

infection, antibiotic use, and genetics in larger studies of HM could demonstrate how these factors impact HM composition and clustering. This study was not powered to evaluate the influence of inflammatory or other specific clinical or demographic parameters, which could influence the composition of HM. Importantly, as birth is an inflammatory event, decreases in the inflammatory patterns of soluble or cellular factors in HM could be in response to a systemic decrease in inflammation after birth. Investigating changes in other bodily fluids, such as blood, saliva, and urine, in parallel to HM could highlight physiologically unique factors to HM. Our comprehensive investigation of HM bioactive factors, including cytokines, growth factors, and immunoglobulins, as well as HM cell types, demonstrated that, during the first two weeks postpartum, many bioactive factors declined in concentration while cellular differentiation in HM increased.

## Methods

**Individuals and sample collection.** Fresh HM specimens from mothers were obtained internally through research coordination at Children's Mercy Kansas City as part of an ongoing study reviewed by Children's Mercy's Institutional Review Board (IRB) in accordance with requirements of local governing regulatory agencies including the Department of Health and Human Services (DHHS) and Food and Drug Administration (FDA) Codes of Federal Regulations, on the Protection of Human Subjects (45 CFR Part 46 and 2l CFR Part 56, respectively).

A total of 37 different individuals were enrolled in this study for HM. Body mass index (BMI) was recorded pre-pregnancy for each individual. Maternal age, infant age, gestational age, maternal BMI, race, and ethnicity were also recorded. With consideration of neonatal nutrition requirements, low volumes of HM were collected at two different timepoints through electronic pumping, week 1 (2–7 days postnatal) and again at week 2 (8–16 days postnatal). Samples were collected during the daytime and processed within 1 h if at room temperature or within 4 h if kept on ice. A total of 16 different females, not involved in milk collection, were enrolled for serum immunoglobulin comparisons.

**Sample processing.** After collecting 3–10 ml fresh milk from the mother, samples were centrifuged at 4 ºC, 800 × g for 15 min to pellet cells. Once cells were pelleted, the soluble non-lipid fraction of the HM was removed and stored at −80 ºC. The cell pellet was then washed three times with cold PBS by centrifugation at 4 ºC, 400 × g for 5 min before immediately proceeding with downstream single-cell library prep.

**Cytokine, chemokine, immune marker, inflammatory, and soluble factor bead-based assays.** After validating serum comparable measures in HM, we utilized magnetic bead-based multiplex assays established on Luminex xMAP technology using the commercially available kits Inflammation 20-Plex (Thermo Fisher, #EPX200-12185-901), Abundant Serum Markers 26-Plex (Thermo Fisher, EPX260-15808-901), and Custom Soluble Factor 10-Plex (Thermo Fisher, #10-PPX). We diluted and processed HM samples according to manufacturer's recommendations for a serum for each kit. Future studies should consider optimizing dilutions for specific analytes based on milk status. Assays were measured for MFI of the following analytes on a Luminex MAGPIX using xPonent: Inflammation 20-Plex: IFN alpha, IFN gamma, IL-1 alpha, IL-1 beta, IL-4, IL-6, IL-8, IL-10, IL-12p70, IL-13, IL-17A (CTLA-8), TNF alpha, IP-10 (CXCL10), MCP-1 (CCL2), MIP-1 alpha (CCL3), MIP-1 beta (CCL4), ICAM-1, CD62E (E-selectin), and CD62P (P-Selectin).

Abundant Serum Markers 26-Plex: NGAL, MMP-3, MMP-9, CD44var (v6), TIMP-1, RBP4, Cystatin C (CST3), PAI-1 (Serpin), SCGF beta, Resistin (ADSF), RANTES (CCL5), CD44, Leptin (LEP), Endoglin (ENG), ZAG, CD14, REG-3a, CD62L (L-selectin), MBL, Adiponectin, Elafin, NRP-1, Osteopontin (OPN), ICAM-1, VCAM-1, and Lactoferrin (LTF).

Custom Soluble Factor 10-Plex: IL-2R, CD30, TWEAK, TNF-RI, TNF-RII, G-CSF, APRIL, BAFF, TRAIL, and CD40L.

**Antibody isotyping.** Antibody heavy chain subclasses IgG1, IgG2, IgG3, IgG4, IgA, IgE, and IgM were determined using the multiplex bead-based Luminex xMAP Technology and an Antibody Isotyping 7-Plex kit (Thermo Fisher, #EPX070-10818-901) following manufacturer recommendations for serum. The assay was measured for MFI on a Luminex MAGPIX using xPonent. Analyte concentrations were determined in xPonent based on automated standard curve interpolations.

**FcR binding assay.** Recombinant human FcR proteins were suspended in 0.1 M sodium bicarbonate at 2 ug/mL and incubated for 2 h at room temperature in 96 well-high binding plates at 100 μL per well. Wells were washed once and 100 μL of

blocking buffer was added per well and incubated overnight at 4 °C. Serum, provided by AMSBio and collected from 16 healthy females aged 24–39 years, and HM was diluted from dilutions ranging from 1:5 to 1:2430, (dilution factor of 3), in the buffer used for blocking. After the overnight blocking, 50 µL of diluted serum or HM was applied to each well and incubated for 1.5 h at room temperature. Following primary incubation, plates were washed twice, and 100 µL of HRP-linked anti-human IgG Fab antibody was applied to each well at a dilution of 1:50,000 in azide-free blocking buffer, and allowed to incubate for 1 h at room temperature. After secondary incubation, plates were washed four times. TMB substrate solution was added at 100 µL per well and incubated for 15 min. About 100 µL of hydrochloric acid stop solution was added immediately after the 15 min TMB incubation. Plates were read at 450 nm within 30 min of applying the acid stop solution.

**Protein concentration**. To assess protein concentration for each sample, we utilized the Pierce BCA Protein Assay (Thermo Fisher, #23225). Bovine serum albumin (BSA) standards were diluted following the kit microplate instructions and performed in duplicate. Samples were diluted at 1:20 and working reagent (WR) at a ratio of 1:8. The assay was measured for absorbance at 562 nm on a BioTek Synergy H1 plate reader. Protein concentrations were interpolated from the standard curve.

**Clustering of samples based on soluble component measurements**. Sample analyte measurements were first normalized as z-scores before employing unsupervised clustering with K of 4 means. Clustering was performed using the Python package scikit-learn (v1.0). Results were graphed using plotly. A volcano plot was generated by performing one-versus-rest T-tests across all four clusters, which provided P values, then adjusted for type 1 error using Bonferroni correction. These adjusted P values were plotted against a log 2 ratio of cluster means.

**Statistics and reproducibility**. All data were normalized using Microsoft Excel; unpaired Mann–Whitney tests were used in Figs. 2, 3a (human milk week 1: $n = 34$, week 2: $n = 26$) with testing for multiple corrected $p$ values conducted using Benjamin–Hochberg FDR at a Q of 1%, and graphing was handled in Graph Pad Prism 9; hierarchical clustering, Pearson correlations, and antibody heatmaps were generated in Morpheus by Broad Institute (RRID:SCR_017386).

**Single-cell RNA sequencing**. Cells from fresh HM were isolated immediately following collection. Cells were counted using a Countess III Automated Cell Counter separated into droplets with a barcoded gel bead using the 10x Chromium instrument (10x Genomics, Pleasanton, CA). Cellular RNA was then reverse transcribed and Illumina adapters were added by following the Chromium Single-Cell 3' v3.3 reagent kit and library protocol (10x Genomics). Samples were single-indexed and sequenced on an Illumina NovaSeq with a run set-up of $28 × 8 × 0 × 94$, aiming for 50,000 reads per cell captured, assuming a 60% capture efficiency.

**Single-cell RNA-seq data analysis**. Sample libraries were not pooled, and instead, library was prepped in separate lanes and sequencing runs. Raw reads from each separately-ran sample were demultiplexed using the Cell Ranger pipeline (v6.0.0). Reads were aligned to the human genome (vGRCH38). The Cell Ranger aggr pipeline was then used to combine data from all 16 samples, and the default method (mapped) was used for normalization. Ranger revealed a post-normalization total of 2,899,096,856 reads at an average of 17,672 reads per cell. There were an estimated 164,049 cells with a median of 735 genes per cell.

Downstream analysis was done with R (v4.0.3) using the Seurat package (v4.0.2). To form the Seurat object, features present in at least three cells were included and cells with >400 and <5000 features were included. Cells with >25% mitochondrial expression were filtered out, reducing the number of cells to a total of 128,016. In pre-processing with Seurat, LogNormalize was used for normalization. The "vst" method, available in "FindVariableFeatures" function included in the Seurat Package, was used to establish the top 2000 variable genes. These top 2000 variable genes were used for data scaling and percentage mitochondrial gene content was used to regress out cell-cell variation in gene expression. To correct for technical batch effects, the Seurat Multi CCA method was used to regress out cell variation. Only the top variable genes across all individual samples were used to renormalize the data, thus excluding variable genes unique to single samples. First, 45 Principal Components (PC) were used to generate Uniform Manifold Approximation and Projection (UMAP) plots and cluster identification was done at resolution 0.6. Default settings were used for the rest of the analysis.

**Cluster annotation strategies**. Gene set enrichment was conducted on differentially expressed genes from Seurat using the reference-based single-cell RNA-seq annotation tool SingleR[17], which used the HPCA (Human Primary Cell Atlas) as a reference to broadly identify cell types. Final cell type labels were established after qualitative assessment using known marker genes in the literature.

**Reporting summary**. Further information on research design is available in the Nature Portfolio Reporting Summary linked to this article.

## Data availability
The RNA sequencing data reported in this paper can be found in the NCBI Sequence Read Archive (SRA) using accession PRJNA835152. All other data is available from the corresponding author on request.

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

## Acknowledgements

We thank all the mothers and serum donors that participated in making this study possible. This work was funded through internal institutional funds from Children's Mercy Research Institute and Children's Mercy Kansas City.

## Author contributions

Conceptualization: T.B., T.L., T.P., and D.M.; sample acquisition: C.L., S.H.P., S.S.E., A.B.S., T.B., T.L., and D.A.L.; laboratory analysis: C.L., S.H.P., and E.S.G.; data analysis: C.L., S.K., and T.B.; writing: C.L., T.B., and R.M.; editing: all authors; funding: T.B.

## Competing interests

The authors declare no competing interests.
