## [Peer Review File · Communications Biology]

Reviewers' comments:

Reviewer #1 (Remarks to the Author):

The authors use several different approaches to characterize HM during the first two weeks postpartum. This is an important and exciting study with datasets that would be of great interest to the HM research field and their work has broad impacts for understanding how early breastfeeding may shape immunity. The major concerns include poor study design and many analyses that do not account for possible co-variates in their datasets, lack of robust analysis of their scRNA-seq data and comparison to several existing datasets, and over interpretation due to use of unadjusted p values at various points in the paper. It is recommended that these major points be addressed in a new version of the paper.

Major Comments:

- The authors state that their significance testing (Mann-Whitney test) was not corrected for multiple comparisons made. The authors should show results for multiple corrected p values and use Benjamini-Hochberg correction as it is more robust. Without correction the data can be overinterpreted. The authors should also elaborate on their statistical tests used in their methods.
- The authors compare binding of IgG subclasses of antibodies to those found in healthy female serum. Since the serum and HM are not matched, it is not reasonable to compare titres between these groups and does not substantiate the claim that IgG antibodies from HM have different binding characteristics. Additionally, the serum profiles of pregnant women and thus those immediately post-birth would be expected to be quite different from healthy control donors. This should be elaborated on and the interpretation adjusted in the paper.
- K means clustering is very sensitive to outliers. It looks like in Figure 1B and in Figure 4A there is one sample that looks very different from the other samples, which could be driving variance on PC1 and overall classification. Is this the same sample in both figures? Did the authors consider removing technical outliers, for example those in clusters 2 and 3, in their dataset before attempting to build a clustering-based classifier? If not, could the authors elaborate on how they determined that these samples are not outliers. How robust is their classifier? For example, could you create a train-test set?
- Can the authors provide any additional information about their donors? For example, route of birth, gestational age, infection status, antibiotic status, BMI, time of milk sampling (e.g. morning or night) or other factors. These co-variates can significantly impact all analyses performed and it is known that infection status significantly impacts the cellular composition of milk. Can the authors comment on this and how this might impact possible outliers for their K means classification and other analyses throughout the paper?
- For analysis of their scRNA-seq data, the authors did not perform any filtering to remove doublet cells. HM is notoriously "sticky" with many enclosed membrane structures that are not cellular in nature. Since the samples were sequenced directly from milk and not sorted using a nuclear or live cell stain prior to performing 10X, doublet removal should be performed on their data before any downstream analysis are performed.
- The authors did not perform any data integration over donor or other batch variables. This should be performed prior to cluster identification to eliminate donor-driven effects in the data. Additionally, when running differential expression the authors should explicitly account for these co-variates.
- The authors claim to identify 25 clusters of cells. They should elaborate on how these clusters were identified and map these clusters to the wealth of existing scRNA-seq datasets that have already delineated HM cell types, such as LC1 and LC2 epithelial cells, in HM. As is, their cell type identification is problematic and unclear. Using existing scRNA-seq reference datasets and established marker genes would better classify their identified cell types.
- The authors use frequency analysis to identify differences in subsets between week 1 and week 2,

but they do not correct for donor which can be a confounding factor in this type of analysis. This should be re-done to explicitly model donor. Additionally, this analysis would make more sense once cell types are properly delineated from their dataset.

- Cell types summarized in Figure 6C are problematic given an earlier comment about cell type identification. Many of these cell types are not expected in HM, so it is unclear what this type of analysis is showing or how robust this is.
- The authors should include UMAPs and bar charts that explore their scRNA-seq data by donor and time post birth. It is unclear if there are donor effects on clustering or sub-clustering analyses that were not properly accounted for during upstream analyses with data integration.
- The point about abundant LTF and other factors that promote infant gut development is well taken and an interesting point. Can the authors elaborate on what cell types may be producing these factors from their dataset or are all of these factors serum derived? Looking at expression of some well-known immunomodulatory factors, like cytokines, chemokines, mucins, and lactoferrins / lysozyme would be interesting in their data.
- The overall paper is well written and references to figures are clear.
- The authors should acknowledge that a major limitation of the study was that most timepoints were analyzed separately given that samples were not paired. This is a major point that should be made clear from the beginning of the paper so as to temper over interpretation of data.
- Can the authors elaborate on and clarify what is meant by this statement: "This work lays the foundation for future data-driven approaches to the categorization of HM and clinically meaningful milk-status markers."

Minor Comments:

- Authors should cite in their introduction newer studies that have determined that the route of vaccination is important for developing immune responses in breast milk, as well as previous work that has characterized many soluble factors in HM. There is a wealth of precedent for factors measured.
- The authors should indicate donors and total numbers of cells included for each scRNA-seq analysis in their methods and their figure legends, particularly for sub-clustering analyses performed on macrophages. A table of cell types identified as a function of donor should be included as a supplemental table.
- The authors claim that a conclusion of their data is that the overall environment may shift from "immune-centric repertoires to HM production as epithelial differentiation increases". This is a reasonably established view in the HM and mammary gland field, thus, the authors should acknowledge previous work in this field and temper any claims of novelty here in their conclusion and rather seek to draw comparisons to what is already known.
- Can the authors investigate expression of other known stem cell markers in Figure S5 and perform cycling analysis to ensure these aren't just cycling cells?
- Can the authors comment on any specific assay format optimization performed for their Ig binding the different FcRs with milk as compared to serum? HM is incredibly rich in proteins and fats, could this impact differences between serum and HM binding phenotypes?
- Can the authors elaborate more on the possible role of SCGFbeta?
- Can the authors make their code for analyses and associated datasets available in public databases prior to publication?

Reviewer #2 (Remarks to the Author):

LeMaster et al. profiled analytes and the transcriptional profile of somatic cells from human milk samples taken during the first two weeks of lactation. The authors generated an interesting and novel data set, profiling an impressive number of samples (37 samples), cells (154,132 cells) and analytes (62 analytes). Through comparing findings from week 1 compared to week 2 of lactation, they have identified shifts in the macrophage populations over this time. I believe this paper and associated dataset will be highly valuable to the scientific community and the authors should be commended for their efforts in generating this manuscript. The statistical analysis is sufficient however a bit more exploratory analysis would add a great deal of value to the paper. I have made a few points below that I hope the authors will consider to maximise the utility of their dataset and findings.

Major points:

1. The scRNA-seq data has not been filtered very stringently, having a 30% cut off for mitochondrial genes is quite high. It is possible that dead/dying cells have been included in the final dataset. In addition, including cells with 200-8500 genes is quite a broad range (Line 196, page 9). Are the authors referring to UMI's here? Or are they referring to number of genes? In any case I believe there is a possibility that empty droplets or doublets may have been included in the analysis. To investigate this possibility, could the authors generate three UMAPs coloured by number of genes (per cell), number of UMI (per cell) and % mitochondrial genes. This would allow the authors and readers to determine whether any identified cluster contains dying cells, doublets or empty droplets which may help determine the identity of the unknown clusters 16 and 24 (Line 221 page 10). If the identity of these cells looks suspicious it might be of value to remove them and make a note of this in the methods.

2. It would be more useful for the authors to combine clusters by cell type/identity rather than keep referring to them as clusters 0-24. Seurat has arbitrarily clustered them based on the expression profiles but not on potential biological function or cell type. The authors have annotated the clusters and thus it is more meaningful for them to refer to the clusters by their cell phenotype. For example, instead of saying clusters 0-4 and that they express epithelial markers, just rename these cells as mammary epithelial cells. This will also be a lot more useful when discussing results, such as in Figure 6B, where comparing changes in week 1 vs week 2 per cell type is more useful than comparing changes by cluster.

3. The authors must take care not to classify cell types based on only 1 or 2 markers, for example the basal cells (Line 213, page 9) and pluripotent (Line 225, page 10) subset. Analysis of canonical markers of KRT14, ACTA2, TP63 and MME should be included and shown to be highly co-expressed to adequately define cluster 23 as basal cells. Similarly, cluster 18 cannot be classified as "pluripotent" based on the expression of a single marker. TOP2A is also not a very well-known pluripotent stem cell maker, therefore please show references for this. Authors should additionally examine the canonical pluripotency transcription factors of SOX2, NANOG, POU5F1 if they want to state that these are pluripotent cells. It should be noted that no other study examining milk cells by scRNA-seq analysis has identified basal or pluripotent cells (thus the authors must provide more evidence to claim this). Another thing to note is that pluripotent stem cells are not the same as hematopoietic stem cells. Therefore, I don't believe that Figure S5A adds much and could be removed as it doesn't provide anything extra.

4. Considering that such a valuable dataset has been generated, it is a shame not to compare data from the analyte and scRNA-seq analysis for samples that have had both measures performed on them. It would be very useful to the readers to correlate analyte values (e.g. IL-1B) with the

corresponding gene expression profile (e.g. ILB1) from the scRNA-seq analysis either by generating a pseudobulk signature of all cells together (per sample) or generating a pseudobulk signature for each sample by specific cell types such as immune or epithelial cells.

Minor points:

1. I believe Figure 1 could be separated into two separate figures as Figure 1A provides more of an overview of the analysis, whereas Figure 1B is actual results from the analyte analysis. I would suggest that if these were to be separated it would be highly useful to complement Figure 1A with a table or chart that shows which samples were used for which kinds of analysis, see attachment for an example.

2. For Figure 1B: Could a bar be added that indicates donor along the x axis? It would be very interesting to see whether someone's analyte profile is similar in week 1 compared to week 2 and as the data is presented now, this is not possible to determine. Performing hierarchical clustering along the x-axis would be additionally very useful to better understand if samples cluster more by participant or by week (this could be added as a supplementary figure).

3. I would not define OPN solely as a hematopoietic stem cell regulator in this context (line 122, page 6). Osteopontin has been well studied in the context of human milk and its functions in lactation should be discussed more than its role as a hematopoietic stem cell regulator (1).

4. Figure 3B, the scale is a bit odd. Is this the best way to represent this?

5. Within the section entitled "Monocytes/macrophages had dynamic phenotypes in early to transitional milk production" (page 10), has the data been subsetted to include only samples from individuals who provided milk for both week 1 and week 2? If not, the differences in the analysis might be due more to biological variations in the week 1 vs week 2 cohorts rather than actual shifts in populations between week 1 and week 2. The authors might consider rerunning this analysis to only include week 1 and week 2 samples from the same individuals.

6. In Line 362-363 page 15, the authors say that their data corroborates findings from previous studies but based on the cell types described this doesn't seem to be the case. Please expand your justification and compare cell types found in your study vs the others, i.e. state whether you are seeing LC1 and LC2 cells and roughly compare proportions of cell subtypes in your study compared to previous studies.

References:

1. B. Christensen, E. S. Sørensen, Structure, function and nutritional potential of milk osteopontin. *International Dairy Journal* 57, 1-6 (2016).

We thank the reviewers for their time and consideration in reviewing our manuscript. With this revision, we have addressed each of their comments and suggestions. Below is a point-by-point response to each reviewer comment.

We believe this has greatly improved our study resulting in a stronger manuscript with increased clarity and scientific precision.

Reviewers' Comments:

Reviewer #1:

Major –

1. The authors state that their significance testing (Mann-Whitney test) was not corrected for multiple comparisons made. The authors should show results for multiple corrected p values and use Benjamini-Hochberg correction as it is more robust. Without correction the data can be overinterpreted. The authors should also elaborate on their statistical tests used in their methods.

RESPONSE: We thank the reviewer for pointing this out and have now rerun the analysis using Benjamini-Hochberg correction for multiple comparisons as suggested. All of the comparisons remained statistically significant after correction, and we have replaced previously uncorrected p-values in the text with Benjamini-Hochberg corrected p-values. This correction can be found in the results section.

Lines 122-132: “Statistical significance testing was Benjamini-Hochberg corrected for multiple comparisons made. The only analyte we measured that was significantly higher in week 2 compared with week 1 was the macrophage modulator OPN (P-value = 0.0079) which is often upregulated at sites of inflammation, proliferation, opsonization, and tissue remodeling (**Figure 2A**) (Jiang and Lonnerdal, 2019). Analytes that are involved in growth, tissue remodeling, innate immunity, inflammation, cell migration, adhesion, and metabolism, as well as total protein had decreased levels in week 2 HM compared to week 1 (**Figure 2A**). Over 22 inflammatory analytes were measured, and only 4 of the 26 analytes with significant changes were related to inflammation: chemokine ligands 3 (CCL3, P-value = 0.0386) and 5 (CCL5, P-value = 0.0102), tumor necrosis factor-related apoptosis-inducing ligand (TRAIL, P-value = 0.0493), and APRIL (P-value = 0.0079) (**Figure 2A**).”

Furthermore, we have added to the methods on statistical tests used in deriving these p values. See line 516.

Original p values and adjusted p values (q values) for Figure 2 are shown in this table.

	p value	q value
CCL3	0.0357	0.03868
NGAL	0.043	0.04472
MMP3	0.0001	0.0002
MMP9	0.0005	0.00076
CD44v6	0.0002	0.00035
TIMP-1	0.0313	0.03538
RBP4	0.0001	0.0002
CST3	0.0001	0.0002
PAI1	0.0001	0.0002
SCGFB	0.0001	0.0002
ADSF	0.0014	0.00202
CCL5	0.0083	0.01028
CD44	0.0001	0.0002
LEP	0.0004	0.00065
ENG	0.0001	0.0002
ZAG	0.0001	0.0002
CD14	0.0001	0.0002
REG3a	0.0001	0.0002
MBL	0.0001	0.0002
ADIPOQ	0.0002	0.00035
PI3	0.0278	0.03285
NRP-1	0.0001	0.0002
OPN	0.0061	0.00793
VCAM-1	0.0001	0.0002
APRIL	0.0059	0.00793
TRAIL	0.0493	0.0493
CD40L	0.0489	0.0489
CD44var6	0.0231	0.0489
TIMP1	0.0408	0.0489
CST3	0.0467	0.0489
CD62L	0.0239	0.0489

2. The authors compare binding of IgG subclasses of antibodies to those found in healthy female serum. Since the serum and HM are not matched, it is not reasonable to compare titers between these groups and does not substantiate the claim that IgG antibodies from HM have different binding characteristics. Additionally, the serum profiles of pregnant women and thus those immediately post-birth would be expected to be quite different from healthy control donors. This should be elaborated on and the interpretation adjusted in the paper.

RESPONSE: We agree that not having matched serum and milk samples is a limitation of our study. Moreover, we agree that the Ig profiles may be influenced during pregnancy and lactation. However, overall changes in blood Fc receptor binding due to lactation or pregnancy have not been previously reported. There has been another study, in the context of SARS-CoV-2 infection/vaccination, that did utilize matched blood and milk samples and demonstrated alterations in milk immunoglobulin Fc receptor binding compared to blood that we observed in our study. We have now added discussion of the limitations of our sampling and the additional references in the discussion section.

Lines 355-362: “One limitation of our samples is that we did not have matched serum samples to the milk samples. Lactating women could have altered serum antibody repertoires that are different than non-lactating women. However, our observation agreed

with another study that identified reduced FcγR2a and FcγR3a binding in HM compared to matched serum samples after SARS-CoV-2 infection (Pullen *et al.*, 2021). Moreover, differences in the Fc binding patterns have also been identified in matched samples examining placental transfer to the fetus (Ateyo *et al.*, 2022; Dolatshahi *et al.*, 2022). Our data and these studies suggest that there are differences in antibody affinity for Fc receptors that are transferred to infants via milk and placenta that could impact neonatal immunity. “

3. K means clustering is very sensitive to outliers. It looks like in Figure 1B and in Figure 4A there is one sample that looks very different from the other samples, which could be driving variance on PC1 and overall classification. Is this the same sample in both figures? Did the authors consider removing technical outliers, for example those in clusters 2 and 3, in their dataset before attempting to build a clustering-based classifier? If not, could the authors elaborate on how they determined that these samples are not outliers. How robust is their classifier? For example, could you create a train-test set?

RESPONSE: We agree that after K-means clustering that the identified cluster 2 only contained a single sample. We believe a larger study with more samples would be needed to determine if there are more individuals that would cluster with this sample or if it is a true outlier. As the reviewer points out in our further clustering analysis, this sample clustered separately from all other samples (Cluster 2, Figure 4A) and, as such, was considered an outlier and excluded from representation in Figure 4D. Based on this reviewer’s feedback, we have further removed representation of this sample in Figure 4B and 4C. Furthermore, we have pointed out this sample as a potential outlier in the results.

Lines 168-170: “Cluster 2 had only 1 sample and thus a larger sample size would be required to validate if this was a true cluster or an outlier (Figure 4A).”

Lines 175-176: “We excluded the single sample cluster 2 from analysis.”
We have also added mention of follow-up studies to the discussion section.

Lines 379-381: “Future studies of larger sample sizes and mothers with increased demographic and health diversity could confirm and expand the clustering analysis and features we identified using soluble factors in the milk.”

Due to the smaller size of the dataset, we did not employ a train-test classification.

4. Can the authors provide any additional information about their donors? For example, route of birth, gestational age, infection status, antibiotic status, BMI, time of milk sampling (e.g. morning or night) or other factors. These co-variables can significantly impact all analyses performed and it is known that infection status significantly impacts the cellular composition of milk. Can the authors comment on this and how this might impact possible outliers for their K means classification and other analyses throughout the paper?

RESPONSE: For this study we were able to collect maternal age, infant age, gestational age, maternal BMI, race and ethnicity. These characteristics are reported in table S1. All of the milk sampling was collected during the daytime. We have now added the collection time to the methods section. There are many other factors that the reviewer pointed out that could impact the composition of human milk, but we did not have all of that information recorded for this study. We agree that other clinical and demographic factors, such as infection or genetics, could potentially influence K means classification but that data was not collected for these samples. Moreover, we did not attempt to compare the composition of HM based on those parameters. We appreciate the reviewer's feedback and have added discussion of these limitations and other limitations of our study in the discussion section of the manuscript.

Lines 422-425: "Additionally controlling for other covariates such as infection, antibiotic use and genetics in larger studies of HM could demonstrate how these factors impact HM composition and clustering. This study was not powered to evaluate the influence of inflammatory or other specific clinical or demographic parameters, which could influence the composition of HM."

5. For analysis of their scRNA-seq data, the authors did not perform any filtering to remove doublet cells. HM is notoriously "sticky" with many enclosed membrane structures that are not cellular in nature. Since the samples were sequenced directly from milk and not sorted using a nuclear or live cell stain prior to performing 10X, doublet removal should be performed on their data before any downstream analysis are performed.

RESPONSE: We appreciate the reviewer's observations here. During processing of the cellular components of the HM, cell pellets were washed with PBS multiple times and single-cell suspensions in PBS, as well as cell viability, were confirmed using microscopy. We did not observe any stickiness or abnormalities in our single-cell suspensions before proceeding to scRNA-seq capture. Typically, doublets or multiple cells in an oil droplet occur at a low rate. These cells have more RNA content and usually cluster separately because of the random cells that form in a single droplet. 10X genomics estimates a 7.6% doublet rate using their technology. We further use filtering of the number of genes and counts to remove doublets through CellRanger and Seurat analysis. Below we have attached three UMAPs colored by number of features per cell, number of UMI per cell, and % mitochondrial genes per cell. Areas where low nFeature are present could indicate dying/dead cells, whereas areas of high nFeature could indicate doublets. In Seurat filtering, we utilized cells containing $nFeature_RNA > 200$, $nFeature_RNA < 8500$, and $percent.mt < 30$. Although we cannot accurately identify or eliminate all cell doublets, there was not a distinct cellular cluster identified that could be doublets and the doublets would occur at even random rates across samples so we would not anticipate them to dramatically impact our result analysis and conclusions.

6. The authors did not perform any data integration over donor or other batch variables. This should be performed prior to cluster identification to eliminate donor-driven effects in the data. Additionally, when running differential expression the authors should explicitly account for these co-variates.

RESPONSE: The data here was integrated prior to cluster identification using aggregation function (CellRanger Aggr), a function used to create a single matrix of cell barcodes and gene counts for the groups. During the process each library is normalized for mapped sequencing depth. In order to control for variation in the number of reads per sample (sequencing depth), reads are subsampled from higher-depth libraries until they all have an equal number of reads per cell that are confidently mapped to the transcriptome. We have now added description of these normalization and batch correction to the methods section on single cell RNA-seq data analysis.

Lines 539-542: “To correct for any batch effects, we used the Seurat analysis pipeline Multi CCA method to regress out cell-cell variation in gene expression in order to control for technical variation. The union of variable genes across all individual samples are then utilized to renormalize the data.”

These bioinformatic approaches control for differences in sequencing depth between libraries and attempt to control for any batch effects. Moreover, our goal when we compared gene expression differences was to identify group, and not individual donor, differences. We grouped all samples together and do not do single sample comparisons (eg. all week 1 samples vs. all week 2 samples). Thus, any donor-specific or sample specific genes would not reach statistical significance because of the variability in expression. We did the analysis this way to attempt to eliminate, even true, individual donor variation in our analysis.

7. The authors claim to identify 25 clusters of cells. They should elaborate on how these clusters were identified and map these clusters to the wealth of existing scRNA-seq datasets that have already delineated HM cell types, such as LC1 and LC2 epithelial cells, in HM. As is, their cell type identification is problematic and unclear. Using existing scRNA-seq reference datasets and established marker genes would better classify their identified cell types.

RESPONSE: Unbiased clustering was identified using graph-based clustering methods in the Seurat R package. This was a way to identify clusters based solely on transcriptome

expression using those machine learning methods that do not rely on prior knowledge of cell populations. To match these clusters to known cell types we used existing scRNA-seq reference datasets such as the human primary cell atlas (HPCA) used in SingleR analysis, and the dataset from PanglaoDB used with Enrichr analysis, were utilized for broad cell identifications. Final elucidation was achieved using cell markers previously identified in breast milk literature. We did identify cell populations that were previously described in prior publications such as LC1 and LC2 in our dataset. We have now incorporated these subsets into our results and figures. We have now defined mammary epithelial cells using milk synthesis gene (*LALBA*), LC1 genes (Claudin 4 (*CLDN4*), Krueppel-like factor 6 (*KLF6*)), and LC2 genes (Xanthine Dehydrogenase (*XDH*) and Casein Alpha S1 (*CSN1S1*)), in accordance with previous studies.

This is reflected in Figure 5 and in the results section.

Lines 208-216: “In agreement with previous studies, the most abundant cells were mammary epithelial cells, which represented 59% of total cells across nine clusters (Twigger *et al.*, 2022; Nyquist *et al.*, 2022) (**Table S4**). These mammary epithelial clusters were enriched with milk synthesis gene lactalbumin (*LALBA*) (**Figures 5B, 5C, S4A and S4B**). Mammary epithelial clusters showed high expression of luminal mammary cell (LC) genes, described in previous studies, with LC1 markers (Claudin 4 (*CLDN4*) and Krueppel-like factor 6 (*KLF6*)) highlighting two clusters (Twigger *et al.*, 2022; Nyquist *et al.*, 2022) (**Figure 5A, 5B and 5D**). Conversely, LC2 markers (Xanthine Dehydrogenase (*XDH*), Casein Alpha S1 (*CSN1S1*)) identified a larger group of cells across 7 clusters and less distinct sets of genes than LC1 clusters (**Figure 5B, 5D and Table S5**).”

In response to this reviewer’s comment as well as comments from other reviewers, we labeled the cells using the cell type assignment rather than referring to specific graph-based clustering numbers (starting with Figure 5B). Although graph-based clustering identified different subclusters in specific cell types, the biological importance of these distinctions are unclear. We hope the relabeling assists the readers with interpretation of our results based on canonical cell type identification.

8. The authors use frequency analysis to identify differences in subsets between week 1 and week 2, but they do not correct for donor which can be a confounding factor in this type of analysis. This should be re-done to explicitly model donor. Additionally, this analysis would make more sense once cell types are properly delineated from their dataset.

RESPONSE: We thank the reviewer for this feedback (with comment 9 and 10) and have removed Figure 6C and replaced Figure 6B with a bar graph of the frequency of each cell type (represented across all clusters) as they are related to donors at each week. Additionally, we have provided donor specific cell counts and frequencies per delineated cell type/timepoint in revised Table S4.

9. Cell types summarized in Figure 6C are problematic given an earlier comment about cell type identification. Many of these cell types are not expected in HM, so it is unclear what this type of analysis is showing or how robust this is.

RESPONSE: We agree and have removed this figure from the manuscript and revised Figure 6B as described above.

10. The authors should include UMAPs and bar charts that explore their scRNA-seq data by donor and time post birth. It is unclear if there are donor effects on clustering or sub-clustering analyses that were not properly accounted for during upstream analyses with data integration.

RESPONSE: We have now integrated the frequency analysis not based on cluster (which we did observe some donor specific responses) but based on cell type to control for donor-specific clusters as Figure 6B. We have included the donor specific data for each cluster as Table S4.

11. The point about abundant LTF and other factors that promote infant gut development is well taken and an interesting point. Can the authors elaborate on what cell types may be producing these factors from their dataset or are all of these factors serum derived? Looking at expression of some well-known immunomodulatory factors, like cytokines, chemokines, mucins, and lactoferrins / lysozyme would be interesting in their data.

RESPONSE: We appreciate the reviewer's feedback on this observation and have created a new figure (Figure 8) to examine the expression levels and cell types that express the most abundant soluble factors we identified in the milk. Feature plots that show the gene expression in the single-cell dataset of the top 5 most abundant soluble analytes that we analyzed (LTF, CD14, APRIL, OPN, and ICAM-1) are included. Feature plots for genes *LTF*, *CD14*, and *OPN* showed abundant expression throughout epithelial and monocyte/macrophage cells. Interestingly however, *ICAM1* and *TNFSF13* (APRIL) genes were widely distributed but not highly expressed across clusters, suggesting that the high content found in human milk may be stored differently or produced elsewhere. We have added a short results section that corresponds to Figure 8.

Lines 275-293: ***“Expression of genes that encode abundant soluble protein factors in HM detected in the cellular compartment.*** We sought to determine the gene expression patterns within single cells of the most abundant soluble proteins that we identified in human milk. The five most abundant soluble protein factors based on median concentration levels among all the HM samples were LTF, CD14, OPN (encoded by *SPP1* gene), APRIL (encoded by *TNFSF13*) and ICAM-1 (**Figure 1**). We determined the gene expression pattern of each of the genes that encode these soluble factors across all the HM cell populations (**Figure 8**). All five genes were detected in the HM cells scRNA-seq. *LTF*, *SPP1*, and *CD14* had the highest expression levels across the HM cells, whereas *TNFSF13* and *ICAM1* had lower expression (**Figure 8**). *LTF*, *SPP1*, and *CD14* were most frequently detected in the mammary epithelial cells (66%, 62%, and 58% respectively), but also were detected in monocyte/macrophage (23%, 29%, and 36% respectively) and other immune cell populations (**Figure 8**). *TNFSF13*, was most frequently expressed in monocyte/macrophage cell types (49%), but also highly abundant in the mammary epithelial cells (47%). *ICAM1* expression was more frequently detected

in the immune cell populations (monocyte/macrophage, neutrophil and B cells; 88%) with smaller frequency of expression in the mammary epithelial cell populations (12%), although still detected in those cells (**Figure 8**). This data suggested that HM cells could express the proteins that are detected in the soluble HM fraction, and some factors/genes have a cell-type specific pattern of expression.”

12. The overall paper is well written and references to figures are clear.

RESPONSE: We appreciate the reviewer’s compliment.

13. The authors should acknowledge that a major limitation of the study was that most timepoints were analyzed separately given that samples were not paired. This is a major point that should be made clear from the beginning of the paper so as to temper over interpretation of data.

RESPONSE: We appreciate the reviewer’s suggestion and have added clarity to this in the introduction.

Lines 82-84: “We also used scRNA-seq analysis, as a method of elucidating transcriptional profiles of aggregated HM cells at week 1 and week 2, separately. Not all samples analyzed at week 1 and week 2 were paired”

14. Can the authors elaborate on and clarify what is meant by this statement: “This work lays the foundation for future data-driven approaches to the categorization of HM and clinically meaningful milk-status markers.”

RESPONSE: We have removed this sentence from the manuscript.

Minor –

1. Authors should cite in their introduction newer studies that have determined that the route of vaccination is important for developing immune responses in breast milk, as well as previous work that has characterized many soluble factors in HM. There is a wealth of precedent for factors measured.

RESPONSE: Three more considerable citations have been added on the subject of infant protection via maternal vaccination on line 59 (Zheng *et al.*, 2022; Atyeo *et al.*, 2022; Saso and Kampmann, 2020), two more citations on line 49 (Garofalo, 2010; Ferrari *et al.*, 2020) and one on line 65 (Kielbasa *et al.*, 2021) for the soluble factors analyzed.

2. The authors should indicate donors and total numbers of cells included for each scRNA-seq analysis in their methods and their figure legends, particularly for sub-clustering analyses performed on macrophages. A table of cell types identified as a function of donor should be included as a supplemental table.

RESPONSE: See response to major comment 8. Table S4 now includes the number of cells from each donor. Additionally, a new supplemental table (Table S8) reflects the number of macrophage cells from each donor as well as their total cells.

3. The authors claim that a conclusion of their data is that the overall environment may shift from “immune-centric repertoires to HM production as epithelial differentiation increases”. This is a reasonably established view in the HM and mammary gland field, thus, the authors should acknowledge previous work in this field and temper any claims of novelty here in their conclusion and rather seek to draw comparisons to what is already known.

RESPONSE: The authors have removed and tempered claims of novelty here.

4. Can the authors investigate expression of other known stem cell markers in Figure S5 and perform cycling analysis to ensure these aren’t just cycling cells?

RESPONSE: The authors have removed Figure S5, a sub-clustering of CD34+ cells, and while we had explored other known markers for stem cells such as *OCT4 (POU5F1)*, *SOX2*, and *NANOG*, *SOX2* transcripts were not found in our dataset and *OCT4* and *NANOG* were found in low abundance. Their violin plots are attached here for visualization.

We added these findings to the results.

Lines 231-235: “We further investigated hematopoietic stem cells (HSC) in the HM cellular population and found 76 cells with HSC marker cluster of differentiation 34 (*CD34*). Additionally, embryonic stem cell genes POU Class 5 Homeobox 1 (*POU5F1*), Homeobox protein NANOG (*NANOG*), and SRY-box 2 (*SOX2*) revealed 44, 27, and 0 cells, respectively.”

Furthermore, cluster 18 was previously identified as pluripotent but we have tempered this claim and re-named this cluster for cell-cycling.

5. Can the authors comment on any specific assay format optimization performed for their Ig binding the different FcRs with milk as compared to serum? HM is incredibly

rich in proteins and fats, could this impact differences between serum and HM binding phenotypes?

RESPONSE: We appreciate the reviewer's feedback. We performed ELISA and bead-based assays on multiple proteins and saw no observable technical challenges between serum and human milk. However, the reviewer makes a good point about the rich nature of milk and we have added this consideration to the discussion.

Lines 364-366: "Additionally, processing and optimization strategies should be employed to consider the high fat and protein content of milk, when compared to serum, which may or may not impact binding efficiency."

6. Can the authors elaborate more on the possible role of SCGFbeta?

RESPONSE: SCGF β is incompletely understood, especially in the context of human milk, and has mostly been studied in stem cell therapy as a direct measure of hematopoietic stem cells and as a growth factor in tumor cells. We do know that SCGF β is linked to the generation and differentiation of macrophage in concert with other cytokines, but little is known beyond that. One speculation is that due to the sheer number of macrophages in human milk, SCGF β abundance reflects the differentiation of monocytes into macrophage for specialized functions.

7. Can the authors make their code for analyses and associated datasets available in public databases prior to publication?

RESPONSE: We have made the data publicly accessible in the NCBI short read archive (SRA: PRJNA835152) and can provide additional support/analysis on request.

Reviewer #2:

Major –

1. The scRNA-seq data has not been filtered very stringently, having a 30% cut off for mitochondrial genes is quite high. It is possible that dead/dying cells have been included in the final dataset. In addition, including cells with 200-8500 genes is quite a broad range (Line 196, page 9). Are the authors referring to UMI's here? Or are they referring to number of genes? In any case I believe there is a possibility that empty droplets or doublets may have been included in the analysis. To investigate this possibility, could the authors generate three UMAPs coloured by number of genes (per cell), number of UMI (per cell) and % mitochondrial genes. This would allow the authors and readers to determine whether any identified cluster contains dying cells, doublets or empty droplets which may help determine the identity of the unknown clusters 16 and 24 (Line 221 page 10). If the identity of these cells looks suspicious it might be of value to remove them and make a note of this in the methods.

RESPONSE: We thank the reviewer for this feedback and have generated UMAPs for gene count, UMI, and percent mitochondria. While the range for genes may be broad, cells have been shown to express a range of genes depending on the cell type and state.

To control for variation in the number of reads per sample (sequencing depth), reads were subsampled from higher-depth libraries until they all had an equal number of reads per cell that were confidently mapped to the transcriptome. Furthermore, in order to form the Seurat object, and correct for any batch effects, features present in at least 3 cells were included and cells with >200 and <8500 genes were included. While 10X estimates that approximately 7.6% of a 10,000 cell load will result in doublets, with our filtering steps we do not suspect death, droplets, or doublets to be an issue in our clustering.

During the process each library is normalized for mapped sequencing depth. In order to control for variation in the number of reads per sample (sequencing depth), reads are subsampled from higher-depth libraries until they all have an equal number of reads per cell that are confidently mapped to the transcriptome. We have now added description of these normalization and batch correction to the methods section on single cell RNA-seq data analysis.

Lines 539-542: “To correct for any batch effects, we used the Seurat analysis pipeline Multi CCA method to regress out cell-cell variation in gene expression in order to control for technical variation. The union of variable genes across all individual samples are then utilized to renormalize the data.”

2. It would be more useful for the authors to combine clusters by cell type/identity rather than keep referring to them as clusters 0-24. Seurat has arbitrarily clustered them based on the expression profiles but not on potential biological function or cell type. The authors have annotated the clusters and thus it is more meaningful for them to refer to the clusters by their cell phenotype. For example, instead of saying clusters 0-4 and that they express epithelial markers, just rename these cells as mammary epithelial cells. This will also be a lot more useful when discussing results, such as in Figure 6B, where comparing changes in week 1 vs week 2 per cell type is more useful than comparing changes by cluster.

RESPONSE: We appreciate the reviewer’s suggestion and have now relabeled each cluster by cell type starting with Figure 5B. In the future figures/results we have referred to them in by cell type rather than cluster ID and have removed unnecessary numerical cluster references. This includes Figure 6 when comparing week 1 and week 2 cell types as the reviewer suggested.

- The authors must take care not to classify cell types based on only 1 or 2 markers, for example the basal cells (Line 213, page 9) and pluripotent (Line 225, page 10) subset. Analysis of canonical markers of KRT14, ACTA2, TP63 and MME should be included and shown to be highly co-expressed to adequately define cluster 23 as basal cells. Similarly, cluster 18 cannot be classified as “pluripotent” based on the expression of a single marker. TOP2A is also not a very well-known pluripotent stem cell maker, therefore please show references for this. Authors should additionally examine the canonical pluripotency transcription factors of SOX2, NANOG, POU5F1 if they want to state that these are pluripotent cells. It should be noted that no other study examining milk cells by scRNA-seq analysis has identified basal or pluripotent cells (thus the authors must provide more evidence to claim this). Another thing to note is that pluripotent stem cells are not the same as hematopoietic stem cells. Therefore, I don’t believe that Figure S5A adds much and could be removed as it doesn’t provide anything extra.

RESPONSE: We appreciate the reviewer’s suggestions and have excluded Figure S5 from the manuscript to reduce over-interpretation of potential stem cells. In addition to using genes from the literature that are associated with cell types, we utilized existing scRNA-seq reference datasets such as the human primary cell atlas (HPCA) used in SingleR analysis, and the dataset from PanglaoDB, used with Enrichr analysis, for broad cell identifications. Final elucidation was achieved using cell markers previously identified in human milk literature. As such, markers such as KRT14 and KRT5 (basal), were directly selected from previous literature on cells in human milk (Martin Carli *et al.*, 2020; Nyquist *et al.*, 2022). While previous work may not have identified basal cells, it is worth noting that our dataset contains the largest consortium of human milk cells sequenced. However, we have elected to remove our basal cell conclusions. Additionally, we have removed all references to pluripotency and instead defined cluster 18 as cell-cycling (Nyquist *et al.*, 2022). Thus, we chose to show representative genes in feature plots for cells that were also defined through reference datasets.

For reviewer interest about stem and basal cells, we identified a low abundant presence of stem cell transcripts with OCT4 (*POU5F1*) and *NANOG* but failed to find expression of *SOX2*. Violin plots visualizing these cells across timepoints have been attached. Additionally, violin plots of markers for basal cells (*KRT14* and *KRT5*) at each timepoint have been attached.

We added these findings to the results.

Lines 231-235: “We further investigated hematopoietic stem cells (HSC) in the HM cellular population and found 76 cells with HSC marker cluster of differentiation 34 (*CD34*). Additionally, embryonic stem cell genes POU Class 5 Homeobox 1 (*POU5F1*), Homeobox protein NANOG (*NANOG*), and SRY-box 2 (*SOX2*) revealed 44, 27, and 0 cells, respectively.”

4. Considering that such a valuable dataset has been generated, it is a shame not to compare data from the analyte and scRNA-seq analysis for samples that have had both measures performed on them. It would be very useful to the readers to correlate analyte values (e.g. IL-1B) with the corresponding gene expression profile (e.g. ILB1) from the scRNA-seq analysis either by generating a pseudobulk signature of all cells together (per sample) or generating a pseudobulk signature for each sample by specific cell types such as immune or epithelial cells.

RESPONSE: We agree with the reviewer that this would be an interesting integration. Due to the small study size, and the fact that we did not have all the samples individually paired, we did not integrate and analyze the soluble and single-cell expression datasets on a donor basis. However, we did want to include if the most abundant soluble analytes were expressed in the cellular compartment, and if so, in what cell types. We have generated a new Figure 8 that analyzes the expression of the most abundant soluble factors in human milk in the single-cell RNA-seq data. Specifically, in Figure 8, we created feature plots that show the gene expression patterns for the top 5 most abundant soluble analytes that we analyzed (*LTF*, *CD14*, *APRIL*, *OPN*, and *ICAM-1*). Feature plots for genes *LTF*, *CD14*, and *OPN* showed abundant expression throughout epithelial and monocyte/macrophage cells. Interestingly however, *ICAM1* and *TNFSF13* (*APRIL*) genes were widely distributed but not highly expressed across clusters, suggesting that the high content found in human milk may be stored differently or produced elsewhere.

Minor –

1. I believe Figure 1 could be separated into two separate figures as Figure 1A provides more of an overview of the analysis, whereas Figure 1B is actual results from the analyte analysis. I would suggest that if these were to be separated it would be highly

useful to complement Figure 1A with a table or chart that shows which samples were used for which kinds of analysis, see attachment for an example.

RESPONSE: We agree that these specific sample details are important and have included them in Figure S1A to clarify what samples were run at each time point and for each analysis component.

2. For Figure 1B: Could a bar be added that indicates donor along the x axis? It would be very interesting to see whether someone's analyte profile is similar in week 1 compared to week 2 and as the data is presented now, this is not possible to determine. Performing hierarchical clustering along the x-axis would be additionally very useful to better understand if samples cluster more by participant or by week (this could be added as a supplementary figure).

RESPONSE: We appreciate the reviewer's suggestion and have labeled Figure 1B x-axis with sample IDs for each week.

3. I would not define OPN solely as a hematopoietic stem cell regulator in this context (line 122, page 6). Osteopontin has been well studied in the context of human milk and its functions in lactation should be discussed more than its role as a hematopoietic stem cell regulator (1).

RESPONSE: We have removed representation of OPN as a hematopoietic stem cell regulator and clarified primary involvement in inflammation, proliferation, opsonization, and tissue remodeling.

4. Figure 3B, the scale is a bit odd. Is this the best way to represent this?

RESPONSE: The scale presented in 3B is representative of a typical antibody endpoint titer determined using ELISA assays, where a specimen is serially diluted 3-fold and then tested on the ELISA. The endpoint titer is determined by the most diluted specimen that gives positive result above background wells on the ELISA. The bars are visual indicators of the mean value.

5. Within the section entitled "Monocytes/macrophages had dynamic phenotypes in early to transitional milk production" (page 10), has the data been subsetted to include only samples from individuals who provided milk for both week 1 and week 2? If not, the differences in the analysis might be due more to biological variations in the week 1 vs week 2 cohorts rather than actual shifts in populations between week 1 and week 2. The authors might consider rerunning this analysis to only include week 1 and week 2 samples from the same individuals.

RESPONSE: We appreciate the reviewer's feedback. The data has not been subsetted to include or exclude any donors aside from timepoint of collection. After aggregation, samples were analyzed at each week separately. Our goal was not to parallel donors but

instead analyze what cell types are and are not present at each week, separately. This clarification has been added to the introduction text.

Lines 82-84: “We also used scRNA-seq analysis, as a method of elucidating transcriptional profiles of aggregated HM cells at week 1 and week 2, separately.”

Our goal when we compared gene expression differences were to identify group, and not individual donor, differences. We grouped all samples together and do not do single sample comparisons (eg. all week 1 samples vs. all week 2 samples). Thus, any donor-specific or sample specific genes would not reach statistical significance because of the variability in expression. We did the analysis this way to attempt to eliminate, even true, individual donor variation in our analysis.

6. In Line 362-363 page 15, the authors say that their data corroborates findings from previous studies but based on the cell types described this doesn't seem to be the case. Please expand your justification and compare cell types found in your study vs the others, i.e. state whether you are seeing LC1 and LC2 cells and roughly compare proportions of cell subtypes in your study compared to previous studies.

RESPONSE: Existing scRNA-seq reference datasets such as the human primary cell atlas (HPCA) used in SingleR analysis, and the dataset from PanglaoDB used with Enrichr analysis, were utilized for broad cell identifications. Final elucidation was achieved using cell markers previously identified in breast milk literature. We did identify populations of cells in our dataset that have been previously described in other studies, such as LC1 and LC2 epithelial populations. We have now incorporated interpretation of our populations in accordance with these previous studies and added the relevant references. We have now defined mammary epithelial cells using milk synthesis gene (LALBA), LC1 genes (Claudin 4 (*CLDN4*), Krueppel-like factor 6 (*KLF6*)), and LC2 genes (Xanthine Dehydrogenase (*XDH*) and Casein Alpha S1 (*CSN1S1*)), in accordance with previous studies.

This is reflected in Figure 5 and in the results section.

Lines 208-216: “In agreement with previous studies, the most abundant cells were mammary epithelial cells, which represented 59% of total cells across nine clusters (Twigger *et al.*, 2022; Nyquist *et al.*, 2022) (**Table S4**). These mammary epithelial clusters were enriched with milk synthesis gene lactalbumin (*LALBA*) (**Figures 5B, 5C, S4A and S4B**). Mammary epithelial clusters showed high expression of luminal mammary cell (LC) genes, described in previous studies, with LC1 markers (Claudin 4 (*CLDN4*) and Krueppel-like factor 6 (*KLF6*)) highlighting two clusters (Twigger *et al.*, 2022; Nyquist *et al.*, 2022) (**Figure 5A, 5B and 5D**). Conversely, LC2 markers (Xanthine Dehydrogenase (*XDH*), Casein Alpha S1 (*CSN1S1*)) identified a larger group of cells across 7 clusters and less distinct sets of genes than LC1 clusters (**Figure 5B, 5D and Table S5**).”

Reviewers' comments:

Reviewer #1 (Remarks to the Author):

I think the authors have addressed several concerns in their revision and the paper is both well written and the conclusions very exciting! However, there are a few points about scRNA-seq quality and analyses that should be addressed before accepting for publication. The one major outstanding concern is that the quality of the scRNA-seq analysis is not clearly presented in their paper. The authors do not present information about quality metrics in their paper and their responses to initial questions about batch correction and upstream processing are not adequately addressed. Specifically, in their rebuttal they state that they used standard workflows in Cell Ranger to normalize data. Data normalization is distinct from batch correction methods.

- Can the authors confirm that there were no technical batch effects in their data (e.g. by sequencing run, processing batch, etc) and please create a supplemental figure that shows their UMAPs colored by donor and other technical features (nfeatures, %mt)? Also, this added statement doesn't make any sense: "To correct for any batch effects, we used the Seurat analysis pipeline Multi CCA method to regress out cell-cell variation in gene expression in order to control for technical variation. The union of variable genes across all individual samples are then utilized to renormalize the data." What cell-cell variation did the authors regress out? What batch effects were corrected for? Can the authors show that this actually worked? This is also important for their detailed macrophage analyses as well. What is the composition of each of these clusters by donor? By other features in the data? Can the authors add in stacked bar plots to summarize in the supplement for readers. More clarity is needed on the upstream analyses.

- Why do the authors filter for less than 30% mt? This was a comment from the other reviewer as well. Given their UMAP colored by %MT in the rebuttal letter, it looks like this is a significant co-variate that could be driving a lot of their clustering. Several "unique" clusters may be artifacts. Did the authors try adjusting the filtering here or regressing out this signal to make sure it is not impacting their clustering and downstream interpretation?

If the above concerns are addressed and presented clearly in the paper, I have no further reservations and commend the authors on a really exciting paper!

Reviewer #2 (Remarks to the Author):

Many thanks to the authors for taking such care in addressing my and other reviewers' concerns. I wanted to re-emphasise a few additional points however, which I believe are important for the authors to consider.

1. The authors have classified cell types using the following method "Unbiased clustering was identified using graph-based clustering methods in the Seurat R package. This was a way to identify clusters based solely on transcriptome expression using those machine learning methods that do not rely on prior knowledge of cell populations. To match these clusters to known cell types we used existing scRNA-seq reference datasets such as the human primary cell atlas (HPCA) used in SingleR analysis, and the dataset from PanglaoDB used with Enrichr analysis, were utilized for broad cell identifications. Final elucidation was achieved using cell markers previously identified in breast milk literature." (Rebuttal letter, end of page 5). However, in results page 10-12 (including Figure 6 onwards), the author's cell classifications are limited to the general cell subtypes described in the HPCA dataset. It should be noted that using SingleR with the HPCA reference dataset to classify the cells biases the results of these cell types. The original HPCA paper states "Accordingly, a diverse set of human leukocyte gene expression data was collected comprising a total of 1,103 chips from 105

separate studies”¹. If one looks closely at the datasets used to build the HPCA (http://biogps.org/dataset/BDS_00013/primary-cell-atlas/) it is evident that no mammary epithelial cells from breast were included (only mammary fibroblast cells). Similarly, from the PanglaoDB database ([https://panglaodb.se/samples.html?species=human&protocol=all protocols&sort=tissue](https://panglaodb.se/samples.html?species=human&protocol=all%20protocols&sort=tissue)), there is limited data available for human mammary cells from the breast and none from milk. Thus, using these datasets as a benchmark to classify the cells from this data (i.e. simply epithelial) is less accurate than referring to the names of human milk cell subtypes (i.e. LC1 and LC2) that have been previously identified²⁻⁴. Please update the cell type classifications in the results and paper figures for pages 10-12.

2. The authors claim that they have the “largest dataset of HM cells to date” (page 16, lines 391-392). However, I still question the quality of all included cells. The plots included in the rebuttal are a good start, however could the authors please use a log scale and include these as supplementary figures? In page 9, lines 198-199 it is said that “We included cells with >200 but <8,500 expressed genes”. I think the authors are referring to unique molecular identifiers or UMI’s. In Seurat “features” are equal to individual genes and “counts” are individual molecules. If the authors are referring to genes this is really a large number of genes, if they are referring to UMIs then their lower limit is not high enough and risks including empty droplets and not true cells. Indeed, cells can express a large range of numbers of genes but scRNA-seq data from cells in human milk to date has never shown this range, despite similar number of reads per cell sequenced^{3,4}. These factors are very important when deciding the quality of the cells and which cells to include in this dataset. In addition to the UMAPS with logged scale, I would also appreciate to see violin plot showing the distribution of cells per sample for UMIs (count) and genes (features) included as supplementary figures.

Minor points:

1. Page 4, line 72: “The other two studies investigated mammary epithelial cells and their transcriptional signatures in HM” Please remove the word “epithelial” as both these studies also investigated immune cells from human milk.
 2. Page 4, Lines 83-84 “We also used scRNA-seq analysis, as a method of elucidating transcriptional profiles of aggregated HM cells at week 1 and week 2, separately. Not all samples analyzed at week 1 and week 2 were paired.” Would it be more accurate to say “....transcriptional profiles of HM cells from different donors at week 1 and week 2”.
 3. Page 6 line 124-126 “The only analyte we measured that was significantly higher in week 2 compared with week 1 was the macrophage modulator OPN (P-value = 0.0079) which is often upregulated at sites of inflammation, proliferation, opsonization, and tissue remodeling (Figure 2A).” Please also discuss the role of OPN in human milk as this is highly relevant to the findings of the paper and to help readers put these finding in a broader context for the field. This is also the case for page 13, lines 313-314 “OPN is often found up-regulated with other cytokines and is thought to promote monocyte differentiation into macrophage”.
1. As you remove cluster 3 in the discussion in page 8 and are left with cluster 1,2 and 4 would it not make sense to rename the clusters so that the cluster you exclude is called cluster 4?
 2. It would be very helpful for the authors to show a heatmap showing the expression levels for individual cells of the embryonic marker genes. Looking at the supplied violin plots it’s not possible to see whether NANOG and POU5F1 are expressed in the same or different cells. This is vital for interpretation of the data and to help support/not support claims of stem cells in milk.
 3. Page 11, were the cell numbers per sample down sampled to ensure equal numbers of cells were considered per participant? Otherwise, is it possible that the cell changes seen between stage 1 and 2 could be driven by some individual donors who have more cells overall with a dominance of one particular cell type?

4. Page 17, line 404-405 "from the expansion of breast tissue in preparation for milk storage". The breast doesn't really store milk for that long (and the cited reference doesn't discuss this), it is constantly actively synthesising milk even over the course of a breastfeed. Perhaps the authors mean to say "from the expansion of breast tissue in preparation for milk synthesis".
5. Page 20 lines 487 and 492 please replace "uL" with "µL".
6. Page 24, line 563, please specify which data repository the data will be available from and that the data availability is in line with the journals policy (<https://www.nature.com/nature-portfolio/editorial-policies/reporting-standards#availability-of-data>)

References:

- 1 Mabbott, N. A., Baillie, J. K., Brown, H., Freeman, T. C. & Hume, D. A. An expression atlas of human primary cells: inference of gene function from coexpression networks. *BMC Genomics* 14, 632, doi:10.1186/1471-2164-14-632 (2013).
- 2 Martin Carli, J. F., Trahan, G. D. & Rudolph, M. C. Resolving Human Lactation Heterogeneity Using Single Milk-Derived Cells, a Resource at the Ready. *J Mammary Gland Biol Neoplasia* 26, 3-8, doi:10.1007/s10911-021-09489-0 (2021).
- 3 Twigger, A.-J. et al. Transcriptional changes in the mammary gland during lactation revealed by single cell sequencing of cells from human milk. *Nature Communications* 13, 562, doi:10.1038/s41467-021-27895-0 (2022).
- 4 Nyquist, S. K. et al. Cellular and transcriptional diversity over the course of human lactation. *Proc Natl Acad Sci U S A* 119, e2121720119, doi:10.1073/pnas.2121720119 (2022).

Response to referees

We thank the reviewers for their time and consideration in reviewing our updated manuscript. We have modified our manuscript to address the reviewer's comments and suggestions that have led to improved clarity and scientific precision of our study. One major concern raised by both reviewers was the filtering and correction of low-quality cells in the single-cell RNA-seq data. As suggested by the reviewers, we have now applied more stringent filtering and correction (detailed in the responses below) that resulted in filtering of potential low-quality cells, empty droplets or droplets with multiple cells. Despite the removal of these questionable cells, this did not change the overall conclusions of the manuscript, nor did it drastically alter the cell populations identified, but did improve the quality and confidence of the scRNA-seq data. This revised analysis did require regeneration of figures with these cells filtered out that resulted in new versions of main text Figure 5, 6, 7 and 8 along with their supplemental information. With this second revision, we have addressed each of their comments and suggestions. Below is a point-by-point response to each reviewer comment.

Reviewers' Comments:

Reviewer #1:

1. Can the authors confirm that there were no technical batch effects in their data (e.g. by sequencing run, processing batch, etc) and please create a supplemental figure that shows their UMAPs colored by donor and other technical features (nfeatures, %mt)? Also, this added statement doesn't make any sense: "To correct for any batch effects, we used the Seurat analysis pipeline Multi CCA method to regress out cell-cell variation in gene expression in order to control for technical variation. The union of variable genes across all individual samples are then utilized to renormalize the data." What cell-cell variation did the authors regress out? What batch effects were corrected for? Can the authors show that this actually worked? This is also important for their detailed macrophage analyses as well. What is the composition of each of these clusters by donor? By other features in the data? Can the authors add in stacked bar plots to summarize in the supplement for readers. More clarity is needed on the upstream analyses.

RESPONSE: We thank the reviewer for this comment and have clarified points around our attempts to reduce technical batch effects and low-quality cell data. After processing and analysis, we observed that cells clustered by cell types, with overlapping cells from multiple samples. We did not observe cells clustering exclusively by sample that would indicate a batch effect. To illustrate this, we have provided a UMAP plot that is colored by each individual sample and have included this figure (Supplemental Figure S4C). We have also included the cluster composition by sample for all the cells as Supplemental Table S4 and the cluster composition by sample for the macrophage clusters as Supplemental Table S8.

We also would like to point out that optimal cryopreservation protocols or viability data are scarce for cells that are in human milk, we processed each human milk sample fresh, within 4 hours of collection, as was performed in prior studies of human milk. Thus,

there was no pooling or batching of samples for 10X Genomics library preparation or Illumina sequencing. For example, the week 1 samples and week 2 samples were not processed in any batch.

We have also modified the methods section of the manuscript to better describe the steps we have taken to address technical variation and quality control. Briefly, we first, we have utilized aggregation functions in CellRanger that aggregates each sample and normalizes sequencing read depth to reduce any technical noise caused by sequencing depth. This object was then analyzed with Seurat, which includes preprocessing workflows that reduce technical noise and to remove low quality cells using mitochondrial RNA and counts of genes and RNA molecules. During this process, Seurat also uses normalization, scaling and variable feature selection to mitigate technical variation by identifying the most variable genes that are in all the samples and excluding variable genes unique to single samples through the multi CCA method that can correct for batch effects. In response to other comments, we have also applied additional stringency on filtering cell quality and doublets, which can contribute to batch effect, and can be more extensively viewed in the following comment 2 response. The UMAP visualizing distribution of sample-specific cells across clusters is now included in Supplemental Figure S4.

UMAP of cells color-coded by sample ID

2. Why do the authors filter for less than 30% mt? This was a comment from the other reviewer as well. Given their UMAP colored by %MT in the rebuttal letter, it looks like

this is a significant co-variate that could be driving a lot of their clustering. Several “unique” clusters may be artifacts. Did the authors try adjusting the filtering here or regressing out this signal to make sure it is not impacting their clustering and downstream interpretation?

RESPONSE: In our analysis we do regress out the contribution of mitochondrial RNA to further perform graph-based clustering. We have added this aspect to our methods section. Based on reviewer feedback, we have also further increased the quality stringency for cut-off of mitochondrial percentage and gene counts to eliminate potential low quality/stressed cells as well as possible droplets with no cells or multiple cells.

Although there is no standard filter cutoff for mitochondrial percentage, at the time of our initial filtering, there was a single publication that filtered out "Cells [from human milk] with >6,000 unique genes, more than 40,000 unique molecular identifier (UMI) counts, and/or mitochondrial transcript percentages higher than 40" (Martin Carli et al. 2020). Thus, we added stringency by lowering mitochondrial allowance to 30% and adding a floor to unique genes at >200 based on the pattern observed in our data. Since then, other publications have yielded filtering parameters at “>400 genes, >750 UMI, <750 counts, <20% UMIs from mitochondrial genes” (Nyquist et al. 2022); exclusion of cells with “[<]400 genes, doublets ([>]5000 genes), and cells with high mitochondrial fraction (>25%)” (Gleeson et al. 2022); and MAD filtering “cleaned barcodes contained >1000 UMIs and that the percentage of mitochondrial genes compared to overall annotated genes were not higher than 1× the median absolute deviation.” (Twigger et al. 2022). After reviewer feedback, we elected to improve stringency to our filtering and utilized parameters that filtered out cells with **<400 and >5,000 genes, >60,000 UMI, and >25% mitochondrial genes**. This reduced cluster numbers to 24 from 25 and a total cell count from 154,132 to 128,016. We believe this added stringency further improves the quality of the scRNA-seq data and matches the latest human milk scRNA-seq publication’s filtering metrics (Gleeson et al. 2022). These QC comparisons can be visualized below, and the new QC can be seen in Supplemental Figure S4. Additionally, these new filtering parameters have been listed in the methods and results. It is also important to note that this revision, while not changing the narrative or findings, has resulted in some number changes that have been modified throughout the text.

Previous quality control visualizations

Updated quality control visualizations (included in Supplemental Figure S4)

Reviewer #2:

Major –

1. The authors have classified cell types using the following method “Unbiased clustering was identified using graph-based clustering methods in the Seurat R package. This was a way to identify clusters based solely on transcriptome expression using those machine learning methods that do not rely on prior knowledge of cell populations. To match these clusters to known cell types we used existing scRNA-seq reference datasets such as the human primary cell atlas (HPCA) used in SingleR analysis, and the dataset from PanglaoDB used with Enrichr analysis, were utilized for broad cell identifications. Final elucidation was achieved using cell markers previously identified in breast milk literature.” (Rebuttal letter, end of page 5). However, in results page 10-12 (including Figure 6 onwards), the author’s cell classifications are limited to the general cell subtypes described in the HPCA dataset. It should be noted that using SingleR with the HPCA reference dataset to classify the cells biases the results of these cell types. The original HPCA paper states “Accordingly, a diverse set of human leukocyte gene expression data was collected comprising a total of 1,103 chips from 105 separate studies”¹. If one looks closely at the datasets used to build the HPCA (http://biogps.org/dataset/BDS_00013/primary-cell-atlas/) it is evident that no mammary epithelial cells from breast were included (only mammary fibroblast cells). Similarly, from the PanglaoDB database (https://panglaoDB.se/samples.html?species=human&protocol=all_protocols&sort=tissue), there is limited data available for human mammary cells from the breast and none from milk. Thus, using these datasets as a benchmark to classify the cells from this data (i.e. simply epithelial) is less accurate than referring to the names of human milk cell subtypes (i.e. LC1 and LC2) that have been previously identified²⁻⁴. Please update the cell type classifications in the results and paper figures for pages 10-12.

RESPONSE: We thank the reviewer for pointing this out. We have removed GSEA annotation methodologies from supplemental and restricted our annotations to literature-based cell type classifications. We opted to retain the SingleR results in supplemental as we feel that it does contribute to the broadly shifting profile of HM and some readers may find these distinctions useful for their own automated annotation strategies. Throughout the manuscript we now refer to lactocytic epithelial cells as their LC1 or LC2 determinations.

2. The authors claim that they have the “largest dataset of HM cells to date” (page 16, lines 391-392). However, I still question the quality of all included cells. The plots included in the rebuttal are a good start, however could the authors please use a log scale and include these as supplementary figures? In page 9, lines 198-199 it is said that “We included cells with >200 but <8,500 expressed genes”. I think the authors are referring to unique molecular identifiers or UMI’s. In Seurat “features” are equal to individual genes and “counts” are individual molecules. If the authors are referring to genes this is really a large number of genes, if they are referring to UMIs then their lower limit is not high enough and risks including empty droplets and not true cells. Indeed, cells can express a large range of numbers of genes but scRNA-seq data

from cells in human milk to date has never shown this range, despite similar number of reads per cell sequenced. These factors are very important when deciding the quality of the cells and which cells to include in this dataset. In addition to the UMAPS with logged scale, I would also appreciate to see violin plot showing the distribution of cells per sample for UMIs (count) and genes (features) included as supplementary figures.

RESPONSE: First, we have removed the claim of “largest dataset” from the discussion as this statement is not necessary. Second, we applied more stringent filtering for cells in our scRNA-seq as the reviewer suggested to eliminate stressed/low-quality cells as well as remove droplets with multiple cells or that were empty. Please see response to reviewer 1 point number 2 that provides more details on the rationale and specifics for increasing the stringency of the filtering of genes and mitochondrial RNA content. We also provided supplemental data (Supplemental Figure S4) that demonstrates our new quality controls. While, this more stringent approach filtered out potential low-quality cells, it did not alter the overall conclusions of the manuscript. We thank the reviewer for the suggestion as it increased the overall quality and confidence of our scRNA-seq dataset.

Minor –

1. Page 4, line 72: “The other two studies investigated mammary epithelial cells and their transcriptional signatures in HM” Please remove the word “epithelial” as both these studies also investigated immune cells from human milk.

RESPONSE: We have removed the word “epithelial” here.

2. Page 4, Lines 83-84 “We also used scRNA-seq analysis, as a method of elucidating transcriptional profiles of aggregated HM cells at week 1 and week 2, separately. Not all samples analyzed at week 1 and week 2 were paired.” Would it be more accurate to say “....transcriptional profiles of HM cells from different donors at week 1 and week 2”.

RESPONSE: We have modified the text here to reflect the reviewer’s suggestion:
Line 82-84: “We also used scRNA-seq analysis, as a method of elucidating transcriptional profiles of HM cells from different donors at week 1 and week 2.”

3. Page 6 line 124-126 “The only analyte we measured that was significantly higher in week 2 compared with week 1 was the macrophage modulator OPN (P-value = 0.0079) which is often upregulated at sites of inflammation, proliferation, opsonization, and tissue remodeling (Figure 2A).” Please also discuss the role of OPN in human milk as this is highly relevant to the findings of the paper and to help readers put these finding in a broader context for the field. This is also the case for page 13, lines 313-314 “OPN is often found up-regulated with other cytokines and is thought to promote monocyte differentiation into macrophage”.

RESPONSE: We appreciate the reviewer’s feedback here and have added additional references and information on OPN’s importance:

Lines 126-128: “Furthermore, OPN may provide benefits to the infant in the form of immunomodulation, intestinal maturation, anti-inflammatory effects, bone development, neurodevelopment, and prebiotic function (Lonnerdal 2014; Jiang and Lonnerdal 2019; Lund, Giachelli, and Scatena 2009).”

Lines 314-317: “OPN is often found up-regulated with other cytokines and is thought to promote monocyte differentiation into macrophage (Ge et al. 2017). Additionally, OPN may benefit infant development and maturation of gut-associated lymphoid tissue (GALT), central nervous, skeletal, and immune systems (Lund, Giachelli, and Scatena 2009; Jiang and Lonnerdal 2019; Lonnerdal 2014).”

4. As you remove cluster 3 in the discussion in page 8 and are left with cluster 1,2 and 4 would it not make sense to rename the clusters so that the cluster you exclude is called cluster 4?

RESPONSE: We agree with the reviewer and have changed all references in the text, figures, and tables to the excluded cluster as cluster 4.

5. It would be very helpful for the authors to show a heatmap showing the expression levels for individual cells of the embryonic marker genes. Looking at the supplied violin plots it's not possible to see whether NANOG and POU5F1 are expressed in the same or different cells. This is vital for interpretation of the data and to help support/not support claims of stem cells in milk.

RESPONSE: The reviewer provides an important distinction. After analysis, no stem cell markers were co-expressed by the same cell. We have added this declaration to the results.

Lines 234-235: “No cell expressing an embryonic or hematopoietic marker was found co-expressing another stem cell marker.”

6. Page 11, were the cell numbers per sample down sampled to ensure equal numbers of cells were considered per participant? Otherwise, is it possible that the cell changes seen between stage 1 and 2 could be driven by some individual donors who have more cells overall with a dominance of one particular cell type?

RESPONSE: We did not down sample the number of cells as we could risk losing a particular subpopulation or additional cell population frequencies. However, we agree that there are differences in each sample in terms of total number of cells. We have now revised this graph by normalizing the frequency of cells by the total number of cells in each sample. The frequency was first calculated on an individual donor basis, then the frequencies averaged for each cell type and displayed in the graph (Figure 6). Thus, this process “normalized” each frequency based on total number of cells in each sample to control for this variation.

7. Page 17, line 404-405 “from the expansion of breast tissue in preparation for milk storage”. The breast doesn’t really store milk for that long (and the cited reference doesn’t discuss this), it is constantly actively synthesising milk even over the course of a breastfeed. Perhaps the authors mean to say “from the expansion of breast tissue in preparation for milk synthesis”.

RESPONSE: The reviewer has raised a good point and we have adjusted the sentence accordingly

Lines 406-407: “from the expansion of breast tissue in preparation for milk synthesis”

8. Page 20 lines 487 and 492 please replace “uL” with “µL”.

RESPONSE: Instances of “uL” have been replaced with “µL” throughout.

9. Page 24, line 563, please specify which data repository the data will be available from and that the data availability is in line with the journals policy (<https://www.nature.com/nature-portfolio/editorial-policies/reporting-standards#availability-of-data>)

RESPONSE: The data availability statement has been updated as such and is in line with the journal’s policy: “The RNA sequencing data reported in this paper can be found in the NCBI Sequence Read Archive (SRA) using accession PRJNA835152.”

References

- Ge, Q., C. C. Ruan, Y. Ma, X. F. Tang, Q. H. Wu, J. G. Wang, D. L. Zhu, and P. J. Gao. 2017. 'Osteopontin regulates macrophage activation and osteoclast formation in hypertensive patients with vascular calcification', *Sci Rep*, 7: 40253.
- Gleeson, J. P., N. Chaudhary, K. C. Fein, R. Doerfler, P. Hredzak-Showalter, and K. A. Whitehead. 2022. 'Profiling of mature-stage human breast milk cells identifies six unique lactocyte subpopulations', *Sci Adv*, 8: eabm6865.
- Jiang, R., and B. Lonnerdal. 2019. 'Osteopontin in human milk and infant formula affects infant plasma osteopontin concentrations', *Pediatr Res*, 85: 502-05.
- Lonnerdal, B. 2014. 'Infant formula and infant nutrition: bioactive proteins of human milk and implications for composition of infant formulas', *Am J Clin Nutr*, 99: 712S-7S.
- Lund, S. A., C. M. Giachelli, and M. Scatena. 2009. 'The role of osteopontin in inflammatory processes', *J Cell Commun Signal*, 3: 311-22.
- Martin Carli, J. F., G. D. Trahan, K. L. Jones, N. Hirsch, K. P. Rolloff, E. Z. Dunn, J. E. Friedman, L. A. Barbour, T. L. Hernandez, P. S. MacLean, J. Monks, J. L. McManaman, and M. C. Rudolph. 2020. 'Single Cell RNA Sequencing of Human Milk-Derived Cells Reveals Sub-Populations of Mammary Epithelial Cells with Molecular Signatures of Progenitor and Mature States: a Novel, Non-invasive Framework for Investigating Human Lactation Physiology', *J Mammary Gland Biol Neoplasia*, 25: 367-87.
- Nyquist, S. K., P. Gao, T. K. J. Haining, M. R. Retchin, Y. Golan, R. S. Drake, K. Kolb, B. E. Mead, N. Ahituv, M. E. Martinez, A. K. Shalek, B. Berger, and B. A. Goods. 2022.

'Cellular and transcriptional diversity over the course of human lactation', *Proc Natl Acad Sci U S A*, 119: e2121720119.

Twigger, A. J., L. K. Engelbrecht, K. Bach, I. Schultz-Pernice, S. Pensa, J. Stenning, S. Petricca, C. H. Scheel, and W. T. Khaled. 2022. 'Transcriptional changes in the mammary gland during lactation revealed by single cell sequencing of cells from human milk', *Nat Commun*, 13: 562.